# 15-Lipoxygenase promotes resolution of inflammation in lymphedema by controlling T<sub>reg</sub> cell function through IFN-β

A. Zamora [1], M. Nougué[1], L. Verdu[1], E. Balzan[1], T. Draia-Nicolau[1], E. Benuzzi[1], F. Pujol[1], V. Baillif[2], E. Lacazette[1], F. Morfoisse [1], J. Galitzky [1], A. Bouloumié [1], M. Dubourdeau[2], B. Chaput[3], N. Fazilleau [4], J. Malloizel-Delaunay[5], A. Bura-Rivière[5], A. C. Prats [1] & B. Garmy-Susini [1] ✉

Lymphedema (LD) is characterized by the accumulation of interstitial fluid, lipids and inflammatory cell infiltrate in the limb. Here, we find that LD tissues from women who developed LD after breast cancer exhibit an inflamed gene expression profile. Lipidomic analysis reveals decrease in specialized pro-resolving mediators (SPM) generated by the 15-lipoxygenase (15-LO) in LD. In mice, the loss of SPM is associated with an increase in apoptotic regulatory T (T<sub>reg</sub>) cell number. In addition, the selective depletion of 15-LO in the lymphatic endothelium induces an aggravation of LD that can be rescued by Treg cell adoptive transfer or ALOX15-expressing lentivector injections. Mechanistically, exogenous injections of the pro-resolving cytokine IFN−β restores both 15-LO expression and Treg cell number in a mouse model of LD. These results provide evidence that lymphatic 15-LO may represent a therapeutic target for LD by serving as a mediator of T<sub>reg</sub> cell populations to resolve inflammation.

Lymphedema (LD) is characterized by the accumulation of protein-rich interstitial fluid, a significant inflammatory cell infiltrate, and dysregulated regional immune responses[1,2]. Subsequent adipose tissue deposition and fibrosis promote progressive anatomic distortion, loss of function, and chronic inflammation[2]. In response to injury, the acute inflammation timely orchestrating a protective program that is critical for the tissue repair and restoration of homeostasis[3]. It is divided into 2 phases: initiation and resolution. The initiation phase is marked by tissue edema resulting from increased blood flow and microvasculature permeability. Then, the resolution phase promotes tissue repair and regeneration, allowing the return to homeostasis[4–8]. Regulatory T (T<sub>reg</sub>) cell responses represent critical arms of the inflammation resolution response as they improve the apoptotic cell clearance (efferocytosis) and thus increase the resolution[9,10]. Recently,

a distinct population of T<sub>reg</sub> cells has been described in nonlymphoid tissues such as visceral adipose tissue[11]. This unique resident T<sub>reg</sub> cell population possesses a distinct phenotype and specifically expresses the transcription factor PPAR-γ[12]. T<sub>reg</sub> cell frequency and number in visceral AT can be modulated independently of the lymphoid-organ T<sub>reg</sub> cell populations allowing a sustainable localized action in the AT as observed in LD[13,14]. Importantly recruitment of T<sub>reg</sub> cells to inflamed LD tissues suppresses the T<sub>H</sub>1/T<sub>H</sub>2 immune response and limits tissue fibrosis, leading to improved lymphatic function[15,16]. The resolution of inflammation is orchestrated by specialized pro-resolving mediators (SPMs) derived from polyunsaturated fatty acids. Human lipoxygenases (LO) catalyze the stereoselective dioxygenation of polyunsaturated fatty acids (arachidonic acid (AA), DHA, and EPA)[17]. In particular, the 15-LO enzyme is constitutively expressed in

[1]I2MC, Université de Toulouse, Inserm UMR 1297, UT3, Toulouse, France. [2]Ambiotis SAS, Toulouse, France. [3]Service de Chirurgie Plastique et des Brûlés, Centre Hospitalier Universitaire de Toulouse, Toulouse, France. [4]Infinity, Toulouse Institute for Infectious and Inflammatory Diseases, Inserm UMR1291, CNRS UMR5051, University of Toulouse, 31024 Toulouse, France. [5]Service de Médecine Vasculaire, Centre Hospitalier Universitaire de Toulouse, Toulouse, France. ✉e-mail: barbara.garmy-susini@inserm.fr

immune- and endothelial cells, where it contributes to immune modulation[18] by generating SPMs that display chemotactic activities and endothelial activation[19,20].

Exploratory studies support the utility of targeted anti-inflammatory therapy with the non-steroid agent (ketoprofen) in patients with LD[21]. However, given that SPMs are agonists of resolution, they will represent promising therapeutic solutions as they do not give rise to unwanted side effects, such as immunosuppression[22–24]. Our investigations in LD biopsies have disclosed the central role of inflammatory SPMs in the pathogenesis and maintenance of the disease, revealing tissue inflammation as a mechanistic platform for the development of acquired LD[25]. This study identifies a reduction of immune-modulating SPMs generated by the arachidonic acid 15-LO in an LD arm compared to a control arm in the same patient, suggesting this enzyme is a key factor for the development of LD. Using a mouse model of secondary LD, we find that the pharmacological inhibition of the 15-LO aggravates LD and is accompanied by a decrease of $T_{reg}$ cell number in the limb adipose tissue (AT). These results are confirmed in a transgenic mouse model in which the ALOX15 gene is selectively deleted in the lymphatic system (ALOX15$^{lecKO}$) that exhibits an increase of lymphedema and a disruption of the lymphatic network associated with a decrease in PPARγ+$T_{reg}$ cell population. This is reversed by transduction with a 15-LO-overexpressing lentivector or treatment with the pro-resolving cytokine IFN-β. Our work demonstrates the crucial role of lymphatic endothelial-15-LO-induced increase in $T_{reg}$ cell number in the arm adipose tissue, via modulation of pro-resolving lipid mediators. It constitutes a further therapeutic approach for LD, a disease with no curative therapeutic solution.

## Results

### Human LD tissues exhibit an inflammatory profile

To broadly identify gene expression signatures associated with secondary LD, we performed bulk-RNA sequencing on dermolipectomy tissue samples from women who developed LD after breast cancer (Fig. 1a). Four patient biopsies (normal arm and LD arm in each patient) were studied and the differential expression analysis (DEseq) followed by a protein-coding RNA profiling is depicted in Fig. 1b. Similar gene expression regulation patterns were observed in patients (Fig. 1c). PCA analysis showed limited variance per patient (Supplementary Fig. 1a). Analysis of the data led to the identification of 211 mRNAs that are up- (182 genes) or downregulated (29 genes) in arm with LD compared with the control arm (CTL) as shown in the volcano plot (Fig. 1d and Supplementary Data 1). Overall, hierarchical clustering analysis indicates that inflammatory and lipid metabolism-associated genes are specifically modulated in LD (Fig. 1e, f). In particular, we observed a strong upregulation of PTX3 that is expressed in response to inflamed endothelium[26], amphiregulin (AREG) that stimulates VEGFC expression[27], and IL-6, a major pro-inflammatory cytokine (Fig. 1g).

Surprisingly, genes involved in lymphangiogenesis were upregulated including VEGFC and its receptor FLT4 (Fig. 1h). This probably is in line with the increased density of anarchic, leaky, and dysfunctional lymphatics observed in lymphedematous limb by lymphangiography. We observed an upregulation of genes involved in matrix remodeling including A Disintegrin And Metalloproteinase with Thrombospondin Motifs (ADAMTS) and Tenascin C (TNC) (Fig. 1h) and on the opposite, downregulation of genes associated within heparane sulfate (HS2ST1, MAMDC2, FGFR2) (Fig. 1i and Supplementary Fig. 1b).

To confirm the inflammatory status in lymphedema, flow cytometry analysis of dermolipectomy tissue samples was performed in the lymphedematous arm compared to the normal arm of the same patients. The number of macrophages, B cells, and T cells from arm AT were analyzed by flow cytometry (Fig. 1j, k and Supplementary Fig. 1c–e). Among the immune cell populations, we found an accumulation of lymphocytes compared to macrophages (Fig. 1j). Based on two-way ANOVA followed by the Sidak multiple comparison test, the

CD3 + T and the CD4+T cell numbers are statistically significantly higher in patients with lymphoedema (Fig. 1k). No difference was observed on neutrophil number (Supplementary Fig. 1f).

### Lymphedema adipose tissue exhibits a decrease in inflammation resolution specialized lipid mediators (SPMs)

LD promotes a massive accumulation of adipose tissue in the limb that is in part responsible for the stasis of interstitial fluids leading to inflammatory cell accumulation. To understand whether lipid composition could be associated with changes in immune cell recruitment, we performed lipidomic analysis of dermolipectomies from lymphedematous arm compared to non-injured arm from the same patient (Fig. 2a). Differential analysis of arachidonic acid (AA), docosahexaenoic acid (DHA), and eicosapentaenoic acid (EPA) was performed on skin (Fig. 2b) and adipose tissue (Supplementary Fig. 2 and Tables 1–6). We observed an overall decrease in SPMs in both tissues, with a statistical significance in the skin. We found that arachidonic acid (AA)-derived SPMs were strongly reduced in lymphedema, in particular pro-resolving molecules generated by the 15-Lipoxygenase (15-LO) (Fig. 2b, red stars means $p < 0.05$). We observed a decrease in 15-Hydroxyeicosatetraenoic acid (15-HETE), Lipoxin A4 and B4 (LXA4 and LXB4), the major AA-derivatives (Fig. 2c). In contrast, we did not find any difference in SPMs generated by DHA metabolism (Fig. 2b). This was not explained by the overall mRNA or protein expression of 15-LO in lymphedematous tissues (Fig. 2d, e). However, we observed a reduced molecular weight form of 15-LO (Fig. 2e). As 15-LO glycosylation is not required for its catalytic activity[28] we hypothesized that this change in electrophoretic mobility can be attributed to changes from pre-activated ferric form to ferrous inactivated 15-LO. This discordance might be associated with the poor amount of analyzed tissue (μg) compared to the patient's explant (up to 2.5 kg). However, the number of 15-LO-positive cells decreased in lymphedema tissue samples (Fig. 2f, g). More importantly, we observed that 15-LO is expressed in the lymphatic endothelium (Fig. 2h, i).

### 15-LO activity is reduced in a mouse model of LD

To understand the molecular mechanisms underlying the inflammatory process in LD, we performed a time course analysis of the adipose tissue in a mouse model of LD (Supplementary Fig. 3a). In this model based on a mastectomy associated with a brachial and axillary lymphadenectomy, mice exhibited an increased dermal backflow associated with a significant swelling of the limb (Supplementary Fig. 3b, c). In the early stages of LD (2 weeks post-surgery), no significant difference in lipid content was observed (Supplementary Fig. 3d and Tables 7–9). In contrast, in acquired LD (8 weeks post-surgery), we found closely similar patterns as those observed in human tissue samples (Supplementary Fig. 3e and Tables 7–9). In particular, we observed a decrease in AA-derived SPMs generated by 15-LO (15-HETE) whereas no significant difference was found in 17-HDOHE (DHA metabolite)(Supplementary Fig. 3f, g). Also, the DHA metabolites showed inverse differences with RvD1 reduced and RvD2 increased in LD tissues. We observed that changes in pro-resolving content were associated with a decrease in 15-LO mRNA and protein expression (Supplementary Fig. 3h, i). Importantly, the downregulation of 15-LO was also observed in the lymphatic endothelium (Supplementary Fig. 3j, k). As previously observed in human tissue samples, numbers of macrophages, dendritic cells, and CD8 + T cells were not modified in mouse LD skin (Supplementary Fig. 4a–i). In contrast, we observed an increase in CD4+T cell number in the LD limb (Supplementary Fig. 4h, j).

### 15-LO promotes $T_{reg}$ cell survival in lymphedematous tissues

To evaluate the role of 15-LO in LD, we used the PD146176 pharmacological inhibitor (Fig. 3a, b and Supplementary Fig. 5). In the

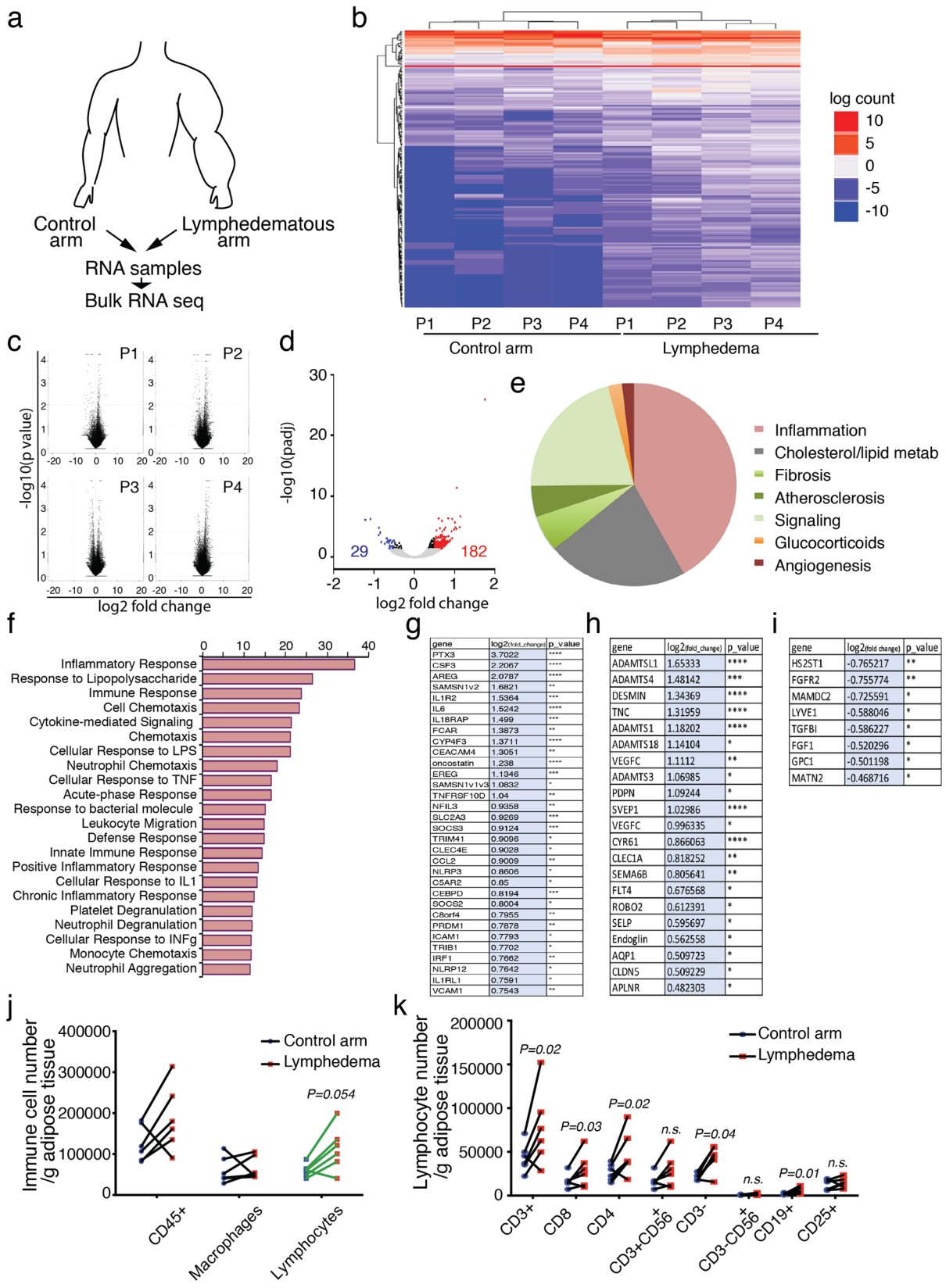

presence of 15-LO inhibitor, LD swelling is increased (Supplementary Fig. 5a) and is associated with dermal fibrosis (Supplementary Fig. 5b, c). An increase in skin thickness was associated with an increase of both dermis (collagen) and hypodermis (AT) accumulation (Supplementary Fig. 5d). Interestingly, the reduction of hair follicles observed in LD was aggravated with PD146176 inhibitor (Supplementary Fig. 5e).

Lymphedema was also characterized by a strong dermal backflow and rerouting of lymphatic vessels (Supplementary Fig. 5f, g). Flow cytometry analysis of the LD AT revealed that 15-LO inhibitor promoted an increase of the CD4+T cell population in the spleen associated with a decrease in $T_{reg}$ cell number (Supplementary Fig. 5h–j). Interestingly, we also found a decrease in the $T_{reg}$ cell population (Foxp3 positive) in

**Fig. 1 | Human lymphedema exhibits an inflammatory profile. a** Schematic representation of the experimental procedure. **b** Heatmap of hierarchical clustering analysis of genes upregulated (red) and downregulated (blue). **c** Volcano plots highlighting the genes that differ between LD/control in each patient. **d** Volcano plot showing the upregulated and downregulated genes in lipodermectomies. **e** Comparison of pathways regulated in LD. **f** Gene ontology of immune regulations in LD. **g** Upregulated expressed genes associated with immune response. **h** Upregulated expressed genes associated with lymphangiogenic response. **i** Downregulated expressed genes associated with lymphangiogenic response

**j** Flow cytometry analysis of adipose tissue from LD dermolipectomies compared to the normal arm on the same patient. **k** Flow cytometry analysis of T cells in LD dermolipectomies. For **b**–**i**, $n = 4$ women with LD (normal arm and LD arm from the same patient). For **j** and **k** $n = 5$ women with LD (tissue biopsies from the normal arm and LD arm from the same patient). Data are shown as mean ± s.e.m.; **g**–**i** Wald test was used to generate $p$-values and log2 fold-changes (*$p < 0.05$, **$p < 0.01$, ***$p < 0.005$, ****$p < 0.001$). **j**, **k** $p$-values are derived from multiple t-tests. Source data are provided as a Source data file.

lymphedematous tissues (Fig. 3a, b). We next investigated the role of these nonlymphoid $T_{reg}$ cells. We took advantage of DEREG transgenic mice that express green fluorescent protein (GFP) protein under control of the endogenous forkhead box P3 promoter/enhancer regions on the transgene[29,30] to perform flow cytometry analysis in lymphedematous AT (Fig. 3c). We found a decrease in GFP-positive Treg cells in LD tissue with ten-fold decrease in $T_{reg}$ cell number in LD limb confirming the loss of this cell population in the injured limb (Fig. 3d, e).

### Lymphatic endothelial-15-LO initiates inflammation resolution in lymphedema

To study selectively the role of lymphatic endothelial-15-LO in the trafficking of $T_{reg}$ cell population in LD, we generated the Prox1CreERT2;Alox15$^{fl/fl}$ conditional knock-out mice in which *Alox15* gene is selectively invalidated in the lymphatic system after tamoxifen induction (ALOX15$^{lecko}$) (Supplementary Fig. 6a–c). In basal condition, the knock down of *Alox15* did not affect lymphatic and blood vessel density, nor body weight (Supplementary Fig. 6d–h). However, we observed a significant aggravation of LD in homozygous and heterozygous mice (Fig. 3f). ALOX15$^{lecko}$ did not exhibit a reduction of lymphatic branching in the LD limb (Fig. 3g, h). However, the dermal limb backflow was increased (Fig. 3i).

Importantly, a strong reduction of the $T_{reg}$ cell population was observed in the limb AT even in the absence of an inflammatory process (Fig. 3j). To confirm the role of Treg cells in LD, Treg adoptive transfer was performed in ALOX15$^{LECKO}$ mice and control littermates (Supplementary Fig. 6h). We observed a significant decrease of lymphedema in both groups (Supplementary Fig. 6i).

To identify if the loss of Treg cells was attributed to an increase in cell death rather than a decrease in cell proliferation, we performed a TUNEL assay and KI67 immunodetection in ALOX$^{15lecko}$ mice. TUNEL staining in control and LD limbs (Fig. 3k) revealed an increase of $T_{reg}$ cell apoptosis in basal condition that was exacerbated in LD condition (Fig. 3l). Notably, $T_{reg}$ cell reduction was related to PPARγ-positive $T_{reg}$ cells, a nonlymphoid population that plays a crucial role in AT homeostasis (Fig. 3m), as shown by immunohistochemistry (Fig. 3m and Supplementary Fig. 7a) and flow cytometry analysis (Supplementary Fig. 7b). Interestingly, despite a slight increase of Treg cells proliferation in LD, ALOX15 depletion had no effect (Supplementary Fig. 7c, d).

### 15-LO lentivector rescue $T_{reg}$ numbers and lymphatic function in LD

We next investigated the therapeutic potential of 15-LO in LD. First, to study the effect of 15-LO as a treatment for LD, we generated a lentivector overexpressing the enzyme. Three intradermal injections of 15-LO lentivector were performed at the time of surgery (Fig. 4a). In this experiment, 15-LO expression was restored in LEC (Fig. 4b) and LD development was completely abrogated using lentivector confirming the crucial role of 15-LO in LD (Fig. 4a). 15-LO lentivector significantly reduced the dermal lymph backflow and the anarchic branching of lymphatic collecting vessels (Fig. 4c, d). We also observed a strong reduction of dermal fibrosis in ALOX15$^{lecKO}$ mice and control littermates (Fig. 4e, f) related to a decrease in both collagen and AT deposition (Fig. 4f, g). This was associated with a rescue of hair follicle

number (Fig. 4h). Interestingly, 15-LO expressing lentivector did not increase the total number of Treg cells (Fig. 4i). However, it significantly improved the rate of Foxp3+PPARγ+ cells confirming the crucial role of this Treg cell subpopulation in LD (Fig. 4j).

To investigate which SPM could mediate the 15LO effect on LD, mice received IP injections Resolvin D1 (RvD1), lipoxin A4 (LxA4), or their receptor inhibitor WRW4 (Fig. 4k and Supplementary Fig. 7e). We did not observe any effect of RvD1, LxA4 and WRW4 on mice swelling (Fig. 4k and Supplementary Fig. 7e). In contrast, 15HETE treatment completely inhibited LD in ALOX15$^{LECKO}$ mice and in control littermates (Fig. 4k). Mechanistically, using in vitro experiments, we found that 15-HETE had no effect on Treg transmigration and adhesion to the lymphatic endothelium (Supplementary Fig. 7f, g), but significantly improved Treg cells survival (Supplementary Fig. 7h).

To investigate whether lymphatic endothelial-15-LO could control T cell trafficking, we knocked down 15-LO in primary cultures of HDLEC using siRNA (Fig. 4l, m). Despite a poor knock down efficiency (30%), it was sufficient to reveal significant downregulation of the chemokine CCL21 in LEC-ALOX15KD (Fig. 4l). We did not find any regulation of SPHK1 and the Sphingosine-1-Phosphate Receptor 1 (S1PR1), whereas SPHK2 that maintains the endothelial integrity was upregulated (Fig. 4n)[31]. The 15-LO knock-down significantly reduced the expression of lymphotoxin beta receptor (LTBR), which plays a crucial role in lymphatic transendothelial $T_{reg}$ cell migration (Fig. 4l)[32]. We also observed an increase in the lymphatic endothelial inflamed status as shown by ICAM1 and VCAM1 upregulation (Fig. 4n).

### IFN-β rescues lymphedema in ALOX15$^{LECKO}$ mice

It was recently shown that IFN-β, produced by pro-resolving macrophages induces IL-10 and 12/15-LO expression in dermal cells[33]. To investigate whether IFN could play a role in LD, we investigated Type I IFN expression in ALOX15$^{LECKO}$ mice (Fig. 5a, b). The knock down of ALOX15 had no effect on IFN-α but reduced IFN-β expression (Fig. 5a, b). Interestingly, the expression was not affected by LD. These results could explain the increase of LD observed in ALOX15$^{LECKO}$ mice compared to control Cre- littermates. We next investigated the effect of recombinant IFN-β in ALOX15$^{LECKO}$ mice (Fig. 5c–j). Mice were injected 3 times per week by 100ng of recombinant IFN-β for 10 days. We observed a rescue of ALOX12/15 gene expression during LD in both ALOX15$^{LECKO}$ mice and control littermates (Fig. 5c). IFN-β treatment significantly reduced dermal backflow and lymphatic rerouting (Fig. 5d, e). This was associated with a strong decrease in limb swelling in ALOX15$^{LECKO}$ mice and control mice (Fig. 5f, g). Importantly, the number of Treg in the lymphedematous limb was rescued and significantly increased in the ALOX15$^{LECKO}$ LD limb (Fig. 5h). Surprisingly, no significant effect on Treg number was found in Cre- control littermates (Fig. 5h). Also, decrease in CCL21 and LTBR expression was found in mice LD tissue compared to uninjured tissue (Fig. 5i, j). However, no significant changes were observed in ALOX15$^{LECKO}$ mice compared to control littermates (Fig. 5i, j). However, when treated with IFN-β, downregulation of CCL21 and LTBR was abolished in both phenotypes (Fig. 5i, j).

## Discussion

Secondary LD represents a major complication of cancer treatment. It is a multifactorial pathology characterized by lymphatic endothelial

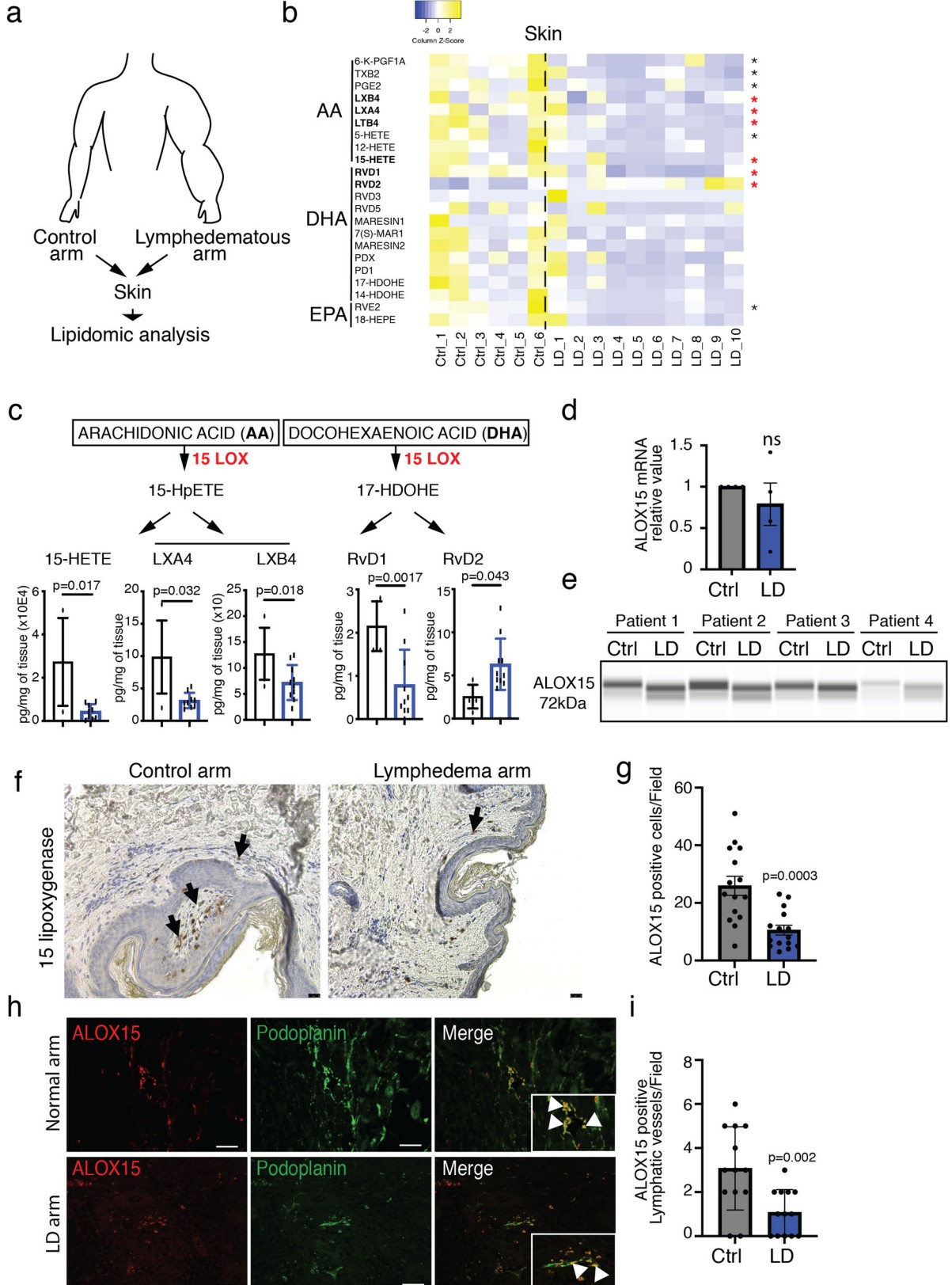

dysfunction, an accumulation of fluid and adipose tissue, a strong dermal fibrosis, and a chronic inflammation in the arm or in the leg. Despite many comorbidities associated with LD, the data of the present study surprisingly demonstrated a reproducible gene expression profile when comparing the lymphedematous arm to the control arm in each patient. Half of these genes are involved in the activation of

inflammation, which is in line with previous studies showing the crucial role of inflammation in lymphedema development. As LD develops months, sometimes years after cancer treatment, it is conceivable to speculate that a low-grade chronic inflammation develops months before physical signs such as skin fibrosis occur. To date, it is well-accepted that anti-inflammatory drugs are the best treatment to some

**Fig. 2 | Downregulation of lipid mediators in human LD. a** Schematic representation of the experimental procedure. **b** Heatmap of lipid mediators derived from arachidonic acid (AA), Docosahexaenoic acid (DHA), and Eicosapentaenoic acid (EPA) analysis in lymphedematous skin. **c** Quantification of 15HETE and 17HDOHE, respectively, AA- and DHA-derived lipid mediators generated by 15-LO conversion. **d** ALOX15 mRNA expression in LD. **e** 15-LO protein expression in LD. **f** Immunodetection of 15-LO in lymphedematous skin (scale bar: 50 μm). **g** Quantification of 15-LO-positive cells in lymphedematous skin.

**h** Immunodetection of 15-LO (red) and podoplanin (green) in normal and lymphedematous skin (Scale bar: 50 μm). **i** Quantification of 15-LO-positive lymphatic vessels in normal and lymphedematous skin. For **b, c** n = 10 women with LD (6 normal and 10 LD arm tissue biopsies). For **d–i** n = 4 women tissue biopsies with LD (normal arm and LD arm from the same patient). Data are shown as mean ± s.e.m.; **b, c** p-values are derived from the Mann–Whitney test. **d, g, i** p-values are derived from unpaired t-tests. Source data are provided as a Source data file.

symptoms of LD. Unresolved inflammation is central to the pathophysiology of common vascular diseases such as atherosclerosis, aneurysm, and deep vein thrombosis[4]. The resolution of vascular inflammation is an important driver of vessel wall remodeling and functional recovery in these clinical settings. In that context, pro-resolving lipid mediators derived from omega-3 polyunsaturated fatty acids orchestrate key cellular processes driving resolution and a return to homeostasis. The identification of their effects on the lymphatic vessels thus arouses great interest in their properties in LD.

Also, a recent study has shown an elevated level of leukotriene B4 (LTB4) in the serum of patients with acquired or primary LD[34]. LTB4 was harmful to lymphatic repair at the concentrations observed in established disease suggesting that LTB4 antagonism is a promising anti-inflammatory drug target for the treatment of acquired LD. The authors also found a decrease in circulating PGE2 showing an imbalance in eicosanoid metabolism. In our study, we performed local dosage by chromatography of eicosanoids in the arm AT. Our results show an overall decrease in eicosanoids and provide a new therapeutic option for LD by using pro-resolving drugs that avoid side effects attributed to anti-inflammatory drugs such as immunosuppression. SPMs are locally synthesized in vascular tissues. They have direct effects on vascular cells-leukocytes interactions and play a protective role in the injury response.

Therefore, they are considered potential vascular therapeutics, as well as candidate biomarkers in vascular disease. Here, we have investigated the molecular and cellular mechanisms involved in the resolution of inflammation in the lymphatic vasculature, to improve tools for clinical measurement, and to better define the potential for "resolution therapeutics" in LD patients. Among the different enzymes that generate lipid mediators, we have identified that the 15-LO is expressed in both immune and lymphatic endothelial cells suggesting an autocrine role in the lymphatic function.

Interestingly, a reduction in the molecular weight of 15-LO was observed by western blot. Although the primary structure contains a number of potential phosphorylation sites there is no evidence that protein phosphorylation/dephosphorylation constitutes a regulatory element of cellular ALOX15 activity[28]. Also, studies have shown that 15-LO glycosylation is not required for its catalytic activity[28]. However, lipoxygenase needs changes from ferrous to ferric species to be activated[35]. Ferrous 15LO is oxidized by Nitric Oxide (NO) to a pre-activated ferric form thus changing the isoelectric point affecting the electrophoretic mobility. These observations favor the hypothesis that changes in SPM concentration in LD can be attributed to a defect in the oxidation status of the enzyme.

As previously described by Krejaschki and colleagues, no deleterious effect of 15-HETE is observed on the lymphatic endothelium compared to the 12-HETE that generates holes in the lymphatic monolayer[36]. In the present study, we do not find any regulation of 12-HETE in LD. However, Krejaschki's study was restricted to a paracrine effect of 12-HETE produced by tumor cells during the metastatic process. Here, we have identified the presence of 15-LO in the lymphatic endothelial cells. Importantly, the expression of 15-LO is correlated with a decrease in Treg, in particular Foxp3⁺CD4⁺ subset infiltration into the lymphedematous adipose tissue. After the adoptive transfer of Treg cells in ALOX15^LECKO mice, we observed a significant decrease in lymphedema even if the transplanted cells were isolated from the spleen. We

postulated that transplanted Treg cells acquired an AT-related phenotype. Treg cells accumulate in a variety of nonlymphoid tissues to exert both anti-inflammatory and homeostatic functions[37]. Recently, a novel subset of Treg has been identified in visceral adipose tissue[11,38,39]. They exhibit a distinct transcriptome from those of lymphoid- and non-lymphoid tissues[37,39,40] and their accumulation is dependent on IL-33[38]. However, targeting 15-LO as a human therapeutic target may have certain barriers and challenges associated with it. In particular, due to the complexity of the Arachidonic Acid metabolic pathway that involves multiple enzymes and metabolites that may have unintended consequences on other enzymes. Also, ensuring effective delivery of 15-LO-derived SPM to the specific tissues or cells of interest can be challenging to exert the desired therapeutic effect while minimizing systemic distribution. Despite these barriers, targeting 15-LO remains an area of active research, and ongoing studies continue to explore its therapeutic potential in various diseases. Here, we found that 15-HETE reduces LD. Importantly, using ALOX15^LECKO mice, we have identified the crucial role of the lymphatic endothelial production of 15-HETE as a key regulator in lymphedema. No significant beneficial effect was observed with RvD1 and LxA4. However, we treated the mice systemically. It would be interesting to continue these studies with local treatments at the LD limb level in order to draw conclusions on effectiveness. In parallel, we identified that IFN-β, a recently identified pro-resolving cytokine in skin fibrosis[33], may represent an exciting alternative for LD resolution treatment. These molecules, alone or in combination, also could constitute good candidates for cardiovascular pathologies in which the lymphatic system is damaged or dysfunctional, such as cardiac ischemia, atherosclerosis, or chronic inflammatory diseases[41–43].

Although no significant changes in PMN were observed in lymphedema, we cannot exclude some contribution of macrophages in the resolution phase. In particular, satiated or non-phagocytic macrophages exhibit distinct gene expression profiles involved in tissue repair and express high levels of IFN-β[44]. In particular, resolution phase macrophages express a selective IFN-β-related gene signature in mice models of bacterial inflammation[45]. In this model, treatment with exogenous IFN-β enhanced bacterial clearance, demonstrating that IFN-β produced by resolution phase macrophages is an effector cytokine in resolving bacterial inflammation. Also, IFN-β enhances clearance of apoptotic PMN via efferocytosis, which is essential for the prevention of chronic inflammation and autoimmunity[46]. Therefore, IFN-β may represent a promising therapeutic target to restore the resolution of inflammation in lymphedema.

We found that LD skin fibrosis was associated with a down-regulation of the number of hair follicles. This phenotype was previously described by Butenko and colleagues during acute inflammation in a mice model deficient for atypical chemokine receptor AKCR2[33]. They found that degeneration and loss of hair follicles were associated with an augmented thickness of the collagenous dermis. Importantly, this was abrogated by IFN-β treatment[33]. However, these observations were only associated with the dermis phenotype as LD is also characterized by an increase of hypodermis AT that is not observed in skin inflammatory models.

Also, quantification of CCL21 and LTBR gene expression was performed in IFN-β -treated mice. The downregulation of CCL21 and LTBR observed in both ALOX15^LECKO mice and control littermates was

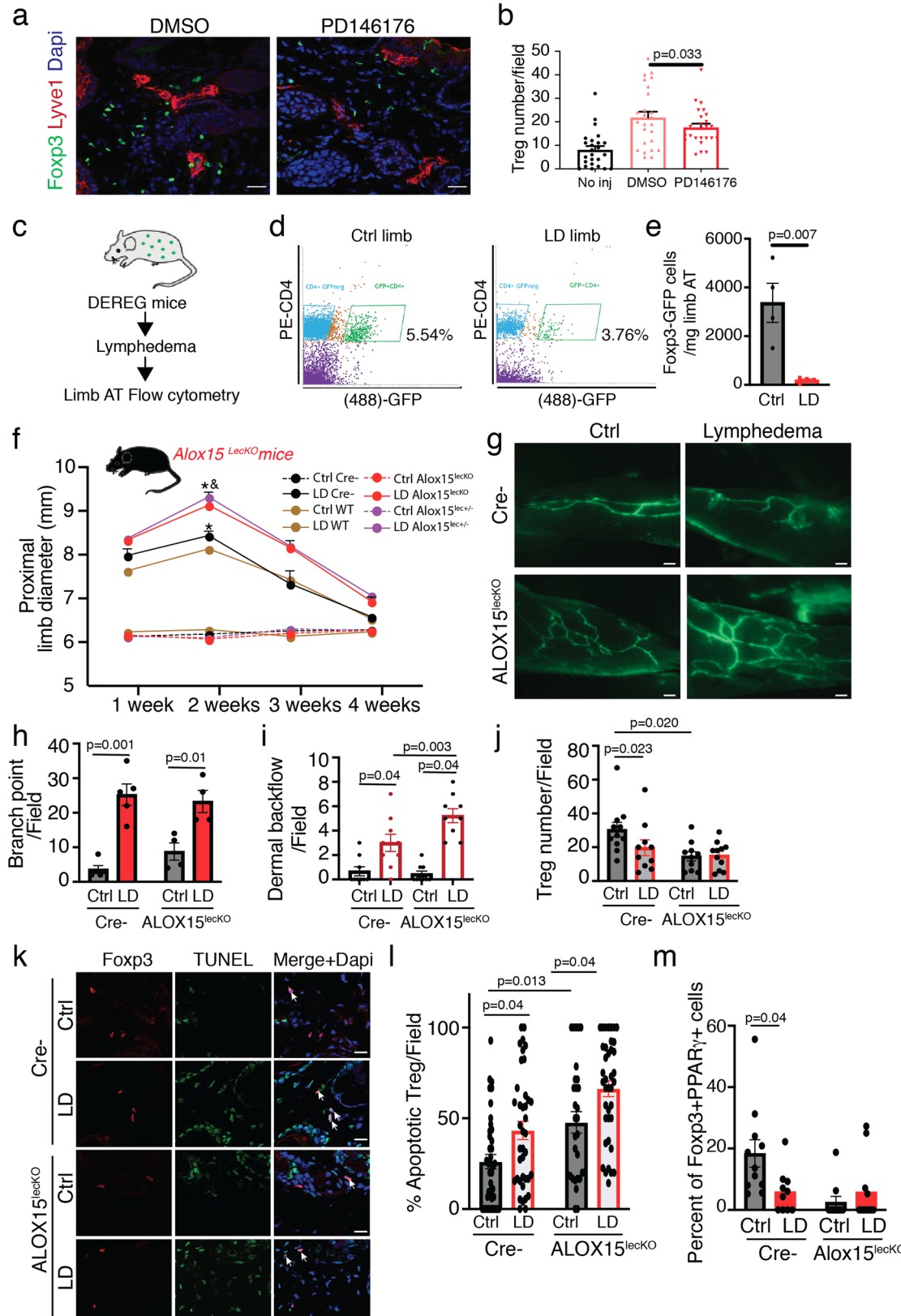

abrogated after IFN-β treatment. This correlates with the phenotype previously observed showing that IFN-β increases CCL21 synthesis by lymphatic endothelial cells[47].

Taken together, these complementary results favor the hypothesis that Treg recruitment initiated by the lymphatic endothelial-15-LO

is a central event in the resolution of inflammation in LD. Importantly, 15-LO does not cause lymphatic damage as observed with 12-LO[36] and thus could represent a potential therapy for LD without promoting any metastatic processes in patients who developed LD after cancer treatment.

**Fig. 3 | Lymphatic endothelial ALOX15 controls T cells function in LD.**
**a** Immunodetection of Foxp3 (green) and Lyve-1 (red) in LD skin (Scale bar: 50 μm).
**b** Quantification of Foxp3-positive cells in LD from mice treated with PD146176.
**c** Schematic representation of experimental procedure in Foxp3-GFP transgenic mice. **d, e** Flow cytometry analysis and quantification of CD4+GFP+ (Foxp3) T$_{reg}$ cell populations in LD adipose tissue. **f** Quantification of the limb diameter in ALOX15$^{LECKO}$ mice. **g** Lymphography of the limb from ALOX15$^{LECKO}$ mice (Scale bar: 1 mm). **h** Quantification of lymphatic branch point in ALOX15$^{LECKO}$ mice. **i** Quantification of lymph dermal backflow in ALOX15$^{LECKO}$ mice. **j** Quantification of the number of T$_{reg}$ cells (Foxp3+) in the limb from ALOX15$^{LECKO}$ mice or control littermates. **k** Immunodetection of apoptotic (TUNEL+) T$_{reg}$ cells (Foxp3+) in the limb from ALOX15$^{LECKO}$ mice (Scale bar: 50 μm). **l** Quantification of apoptotic T$_{reg}$ cells in the limb from ALOX15$^{LECKO}$ mice. **m** Quantification of PPARγ-positive T$_{reg}$ cells in the limb from ALOX15$^{LECKO}$ mice. Data are shown as mean ± s.e.m.; For **b**, $n = 5$ mice per group. For **d**, **e** $n = 4$ mice per group per experiment, 2 independent experiments. For **f**–**i** $n = 4$ mice per group per experiment, 3 independent experiments. **b**, **e** $p$-values are derived from one-way ANOVA. **f**, **h**, **l**, **j**, **l**, **m** $p$-values are derived from Two-way ANOVA. Source data are provided as a Source data file.

## Methods

### Human tissue specimen
Samples were selected as coded specimens under a protocol approved by the INSERM Institutional Review Board (DC-2008-452) and Research State Department (Ministère de la recherche, ARS, CPP2, authorization AC-2008-452). All donors provided written informed consent. When it was possible, some control arm tissue samples were collected for esthetical purposes. In total, 16 lipodermectomy specimens were collected. Samples were obtained from archival paraffin blocks of LD from patients treated at the Rangueil Hospital, Toulouse, France between 2015 and 2016.

### Bulk-RNA sequencing
Total RNA from dermolipectomies was harvested and isolated using the RNeasy mini kit (Qiagen). DNA digestion was performed using the RNase-Free DNase set (Qiagen). Total RNA was then depleted of ribosomal-RNA according to the Eurofins genomics pipeline. RNA sequencing (RNA-Seq) was then performed by Eurofins genomics using paired-end 2 × 150 bp sequencing on Illumina HiSeq. Sequence quality was assessed with FastQC (Galaxy version v0.52) and then trimmed with Trimmgalore v0.3.8.1 with the Galaxy interface. The trimmed reads were then mapped to the Homo sapiens GRCh38.91 reference genome available on ENSEMBL using the STAR aligner v.2.5.2b. The hit counts were summarized and reported using the gene_id feature in the annotation file. The distribution of read counts in libraries was examined before and after normalization. The original read counts were normalized to adjust for various factors such as variations of sequencing yield between samples. These normalized read counts were used to accurately determine differentially expressed genes. After the extraction of gene hit counts, the gene hit counts table was used for downstream differential expression analysis. Using DESeq2 (Galaxy version 0.99.2), a comparison of gene expression between the control arm against LD arm was performed. The Wald test was used to generate $p$-values and log2 fold-changes. Genes with log2FC > 0.5 or log2FC < − 0.5 and an adjusted $p$-value < 0.05 were defined as differentially expressed genes and used for the downstream analysis. The global transcriptional change across the two groups compared was visualized by a volcano plot. Each data point in the volcano plot represents a gene. The log2 fold-change of each gene is represented on the $x$-axis and the log10 of its adjusted $p$-value is on the $y$-axis. Genes with an adjusted $p$-value less than 0.05 and a log2 fold-change greater than 0.5 are indicated by red dots. These represent upregulated genes. Genes with an adjusted $p$-value less than 0.05 and a log2 fold-change less than 0.5 are indicated by blue dots. These represent downregulated genes.

### Gene ontology (GO) analysis
A gene ontology analysis was performed separately on the statistically significant sets of upregulated and downregulated genes, using PANTHER software (version 16.0, http://pantherdb.org/). The Homo Sapiens reference list was used to cluster the set of significantly differentially expressed genes based on their biological processes or pathways and the overrepresentation of gene ontology terms was tested using Fisher exact test.

### Immune cell population profiling in LD
AT from dermolipectomies was digested using type I collagenase (Sigma-Aldrich). Whole AT was sequentially digested with 1:1 dispase (2.4 U/mL in phosphate-buffered saline (PBS), Gibco, 30 min at 37 °C with shaking) and type I collagenase (250 U/mL in PBS 2% bovine serum albumin (BSA), Sigma-Aldrich, 30 min at 37 °C with shaking). After digestion, the cell suspension was filtered through a 250-μm strainer and centrifuged. The erythrocyte lysis step was performed followed by successive filtrations through 100, 70, and 40 -μm strainers. The viable recovered cells were counted and further analyzed by flow cytometry. Immune cells were enriched using a CD45 isolation kit (Stem Cells Technologies) following the manufacturer's protocol. Anti-human FITC-CD4 dilution (1/10), BD Pharmingen, Ref 555346, CLONE RPA-T4 (RUO); PerCP-CD8 dilution (1/10), BD, Ref 345774, CLONE SK1 (CE/IVD); Pe-Cy7-CD56 dilution (1/20); BD Pharmingen, Ref 345774, CLONE B159 (RUO); APC-Cy7-CD19 dilution (1/20); BD Pharmingen, Ref 557791, CLONE SJ25C1 (RUO); APC-CD25 dilution (1/20), BD, ref 340907, CLONE 2A3 (CE/IVD); V450-CD3 diution (1/20); BD Horizon, ref 560365, CLONE UCHT1 (RUO); BV510-CD45 dilution (1/20), BIOLEGEND, ref 304036, CLONE HI30.Flow cytometry was carried out with LSRII BD Fortessa. Cells are expressed as number/g adipose tissue.

**Lipidomic analysis.** Lipids corresponding to 10–50 μg of adipose tissue from the control and LD limb were extracted and an analysis of bioactive lipids was performed. The extraction protocol and liquid chromatography–mass spectrometry (LC-MS/MS) analysis were performed by AMBIOTIS SAS (Toulouse, France) using Standard Operating Procedures adapted from Le Faouder et al.[48]. Briefly, tissues were dilacerated with a scalpel, precisely weighed (about 200 mg), and then crushed with steel beads in methanol (MeOH) using Precellys24 (Bertin Instruments). After protein precipitation and centrifugation, the SPM was extracted from the clear supernatant using an HLB Oasis 96-well extraction plate (Waters). The lipids were finally eluted with methylformate (MeFor) and MeOH. After evaporation of the solvent under N2, the residues were recovered in MeOH/H2O and subjected to LC/MS analysis. Analysis was conducted using a scheduled Multiple Reaction Monitoring mode on a 6500 + QTRAP (Sciex) mass spectrometer equipped with an electrospray ionization source in negative mode. The sample was injected beforehand into the Exion LCAD U-HPLC system (Sciex) and eluted on a KINETEX C18 column (2.1*100 mm; 1.7 μm) with a gradient at 0.5 mL/min of buffer A (H2O, 0.1% formic acid (FA)) and buffer B (MeOH, 0,1% FA) placed into a thermostatic oven at 50 °C.

### Mouse models
All studies received local ethics review board approval and were performed in accordance with the guidelines of the European Convention for the Protection of Vertebrate Animals used for experimentation and according to the INSERM IACUC (France) guidelines for laboratory animal husbandry. All animal experiments were approved by the local branch Inserm Rangueil-Purpan of the Midi-Pyrénées ethics committee,

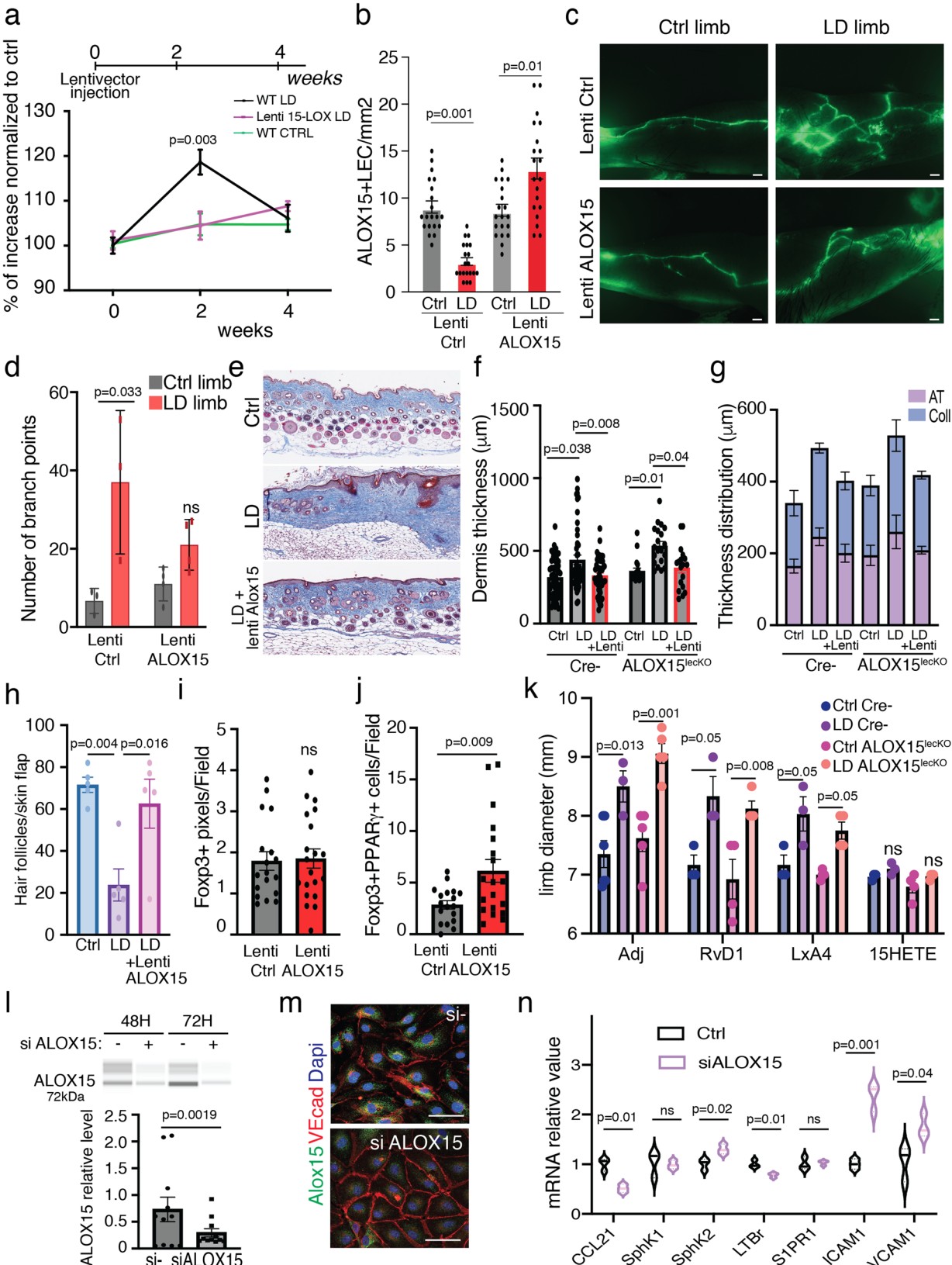

France. Animals from different cages in the same experimental group were selected to assure randomization. Mice were housed in individually ventilated cages in a temperature and light-regulated room in an SPF facility and received food and water *ad libitum*. Female C57BL/6J (6 weeks old) were obtained from Envigo, France.

Prox1CreERT2; Alox15fl/fl mice (C57BL/6J background) were generated by crossing Prox1CreERT2 mice from Dr Makinen's laboratory and ALOX15fl/fl obtained from Jackson laboratory (B6.Cg-Alox15[tm1.1Nadl]/J). DEREG mice were obtained from Jackson laboratory (C57BL/6-Tg (Foxp3-DTR/EGFP) 23.2Spar/Mmjax). *12/15-LO[loxP/loxP]* mice possess *loxP*

**Fig. 4 | 15-LO treatment restores lymphatic function in LD. a** Quantification of the limb diameter in LD from mice treated with three intradermal injections of ALOX15 lentivectors. **b** Quantification of ALOX15-positive lymphatic endothelial cells in the skin from mice treated with intradermal injections of ALOX15 lentivectors. **c** Lymphography of the limb from ALOX15-treated mice (Scale bar: 1 mm). **d** Quantification of lymphatic branch point in ALOX15-treated mice. **e** Masson's trichrome coloration of the lymphedematous skin. **f** Quantification of dermis thickness in mice treated with ALOX15 lentivector. **g** Skin thickness repartition related to collagen (Coll) vs. adipose tissue (AT). **h** Hair follicle quantification in LD skin. **i** Quantification of Foxp3-positive pixels in the limb from mice treated with ALOX15 lentivector. **j** Quantification of the number of PPARγ-positive $T_{reg}$ cells in the limb from mice treated with ALOX15 lentivector. **k** Quantification of the limb diameter in LD from mice treated with Resolvin D1 (RvD1), Lipoxin A4 (LxA4), and 15-HETE. **l** Capillary electrophoresis and its quantification showing ALOX15 knock down in LEC. **m** Immunodetection of 15-LO (green) and VE-Cadherin (red) in LEC (Scale bar: 25 µm). **n** Quantitative RT-qPCR analysis of CCL21, SPHK1, SPHK2, LTBR, S1PR1, ICAM1, and VCAM1 in LEC after ALOX15 knock down. For **a–n** $n = 4$ mice per group, 2 independent experiments. Data are shown as mean ± s.e.m.; (**a, b, d, f, k**) $p$-values are derived from two-way ANOVA. **i, j, l, n** $p$-values are derived from unpaired $t$-tests. Source data are provided as a Source data file.

sites flanking exons 2-5 of the arachidonate 15-lipoxygenase (*Alox15*) gene (C57BL/6J background).

## Mouse model of lymphedema
Lymphedema was established in the left upper limbs of 6-week-old C57Bl/6 female mice. Lymphedema surgical procedure is performed by partial mastectomy of the second mammary gland and is associated with axillary and brachial lymphadenectomy. Limb size was measured using a caliper as previously described[29]. At the end of the experiment, mice euthanasia is performed using a lethal injection of zoletil and xylazine followed by cervical dislocation.

## Microlymphangiographies
Mice were anesthetized with a subcutaneous injection of zoletil (30 mg/kg) and xylazine (10 mg/kg), and the integrity of the lymphatic vasculature of the skin was examined by fluorescence microlymphangiography. A fluorescently labeled macromolecule (70,000 kDa fluorescein isothiocyanate–dextran-conjugated dextran, 2 mg/mL; Sigma-Aldrich, France) was injected into the footpad of the edematous and control leg. Because of its large size, the tracer was taken up by the lymphatics but was excluded from the blood vasculature. As the lymphatic vessels transported fluorescent tracer, it was clearly visible within dermal lymphatic capillaries and collecting vessels, thus providing a clear visualization of lymphatic functionality. Images were acquired using a binocular (Zeiss) and analyzed in Fiji.

## Treg cell population isolation from DEREG mice
After 2 weeks of LD, DEREG mice were sacrificed and their limb AT was collected. AT were ground into a single-cell suspension through a 70 µM cell strainer. Red Blood Cells were removed using the RBC lysis buffer (Abcam) following the manufacturer's instructions. CD4 cells were enriched using a CD4⁺ T Cell isolation kit, mouse (Miltenyi). CD4⁺ T were then stained using Live dead Violet (Invitrogen) to discriminate dead cells and CD4-PE (Biolegend) followed by cell sorting using the BD Influx machine. GFP expressed by the Treg was also used to discriminate and isolate Treg. FACS was carried out with LSRII BD Fortessa.

## Flow cytometry analysis of mouse LD
After 2 weeks of LD, mice were sacrificed, spleens were harvested and cut up finely using scissors, and then incubated with Collagenase A (ROCHE) for 30 at 37 °C, shaking. Digested tissue was processed into single-cell suspensions through a 40-µm cell strainer and then pelleted. Red Blood Cells were removed using the RBC lysis buffer (Abcam) following the manufacturer's instructions. Cells were washed, pelleted, and resuspended in isolation Buffer for CD4 enrichment using CD4⁺ T Cell isolation kit, and mouse (Miltenyi, 130-104-454). It was followed by Live dead violet staining (Invitrogen), CD4-APC-Cy7 dilution (1/50), Biolegend, BLE100526; CD11c-BV605 dilution (1/50), Biolegend 117333; CD206-PerCP/cy5.5 dilution (1/50), Biolegend 141715; F4/80-APC dilution (1/50), Biolegend 123116; CD8-BV510 dilution (1/50), BD BLE100751 stainings before FACS analysis (LSRII BD Fortessa) or Treg adoptive transfer.

Immune cell characterization in mice LD. Skin with adipose tissue was collected from lymphedema or control limb. Three mice were pooled per point. Tissues were digested with Multi Tissue Dissociation Kit 1 (Miltenyi 130-110-201) following the skin program for 3 h on the gentleMACS Dissociator from Miltenyi. Digested tissues were cleared through a cell strainer 100 µm before erythrocyte lysis using Red Blood Cell Lysis Solution (Miltenyi 130-094-183) for 2 min at room temperature. Cell suspensions were obtained after filtration through a cell strainer 40 µm and were resuspended in FACS buffer (PBS, 2% FBS, 0.4% EDTA 2 mM) and then blocked with 10% mouse FcR blocking reagent (Miltenyi) and 2% BSA for 10 min on ice. Cell viability was determined with Live/Dead fixable violet reagent (Invitrogen) in PBS for 20 min on ice before surface antibody staining. Cells were stained for 30 min on ice with CD45-AF488 (Biolegend), CD3-PE-Cy5 (BD Biosciences), CD4-APC-Cy7 (Tonbo), CD8-BV510 (Biolegend), F4/80-APC (Biolegend), CD11b-PE-vio770 (Miltenyi), CD80-AF700 (eBioscience), CD206-PerCP-Cy5.5 (Biolegend), CD11c-BV605 (Biolegend). For Foxp3 intracellular staining, cells were fixed and permeabilized with Foxp3/Transcription Factor Fixation/Permeabilization reagent (eBioscience) for 30 min on ice and stained with Foxp3-PE (eBioscience) for 30 min on ice. Cells were analyzed by FACS Fortessa instrument (BD Biosciences) and data were analyzed using Diva software (BD Biosciences).

## Treg cell adoptive transfer
ALOX15^LECKO mice ($n = 9$) and control littermates were injected with 10,000 GFP+ Treg cells isolated from DEREG mice in the limb AT 3 days post-surgery. Mice were monitored after 14 days.

## Whole-mount immunostaining
Ear skin was dissected from the cartilage layer and fixed in 4% formaldehyde in PBS for 1 h on ice. After washing 3 times in PBS, samples were permeabilized in PBS 0.3% TritonX-100 (PBST) and saturated for 1 h in PBS containing 0.3% TritonX-100 and 3% milk (PBSMT) at RT. After washing 3 times in PBS, samples were incubated with primary antibodies: goat anti-mouse Lyve 1 (R&D) at 1:200; and rat anti-mouse CD31 (BD) at 1:100 in PBST overnight at 4 °C. Secondary antibodies (Jackson ImmunoResearch; 1:300) were incubated for 2 h RT. The samples were mounted on a slide in Fluorescent Mounting Media (Dako). Images were acquired on a DMi8 wide field microscope (Leica) using a Leica DFC9000GT camera. For quantification measurements, at least three regions in the periphery and center of each outer ear skin were analyzed. The threshold level was set to minimize background noise, and then the pixel intensity values for the positive-stained area were measured using NIH ImageJ software (NIH).

## Chemical and reagents
Rabbit anti-mouse LYVE-1 antibody was from Fitzgerald (Fitzgerald, 70R-LR004 1/200) mouse anti-human podoplanin (D2/40) from DAKO (DAKO, 322M-17, 1/50). Anti-human 15 LO (119774, 1/50) and anti-mouse 15-LO (244205 1/50) were from Abcam. Anti-mouse Foxp3 was from Abcam (Ab20034, 1/100). Anti-PPARγ was from Cell signaling (C26H12 1/50). Anti-human VE-Cadherin was from Santa Cruz (sc6458).

## Specialized lipid mediator treatment
ALOX15^LECKO mice and control littermates were injected IP with 100 ng RvD1 (Cayman), 100 ng LxA4 (Cayman) or 300 ng 15-HETE (Cayman)

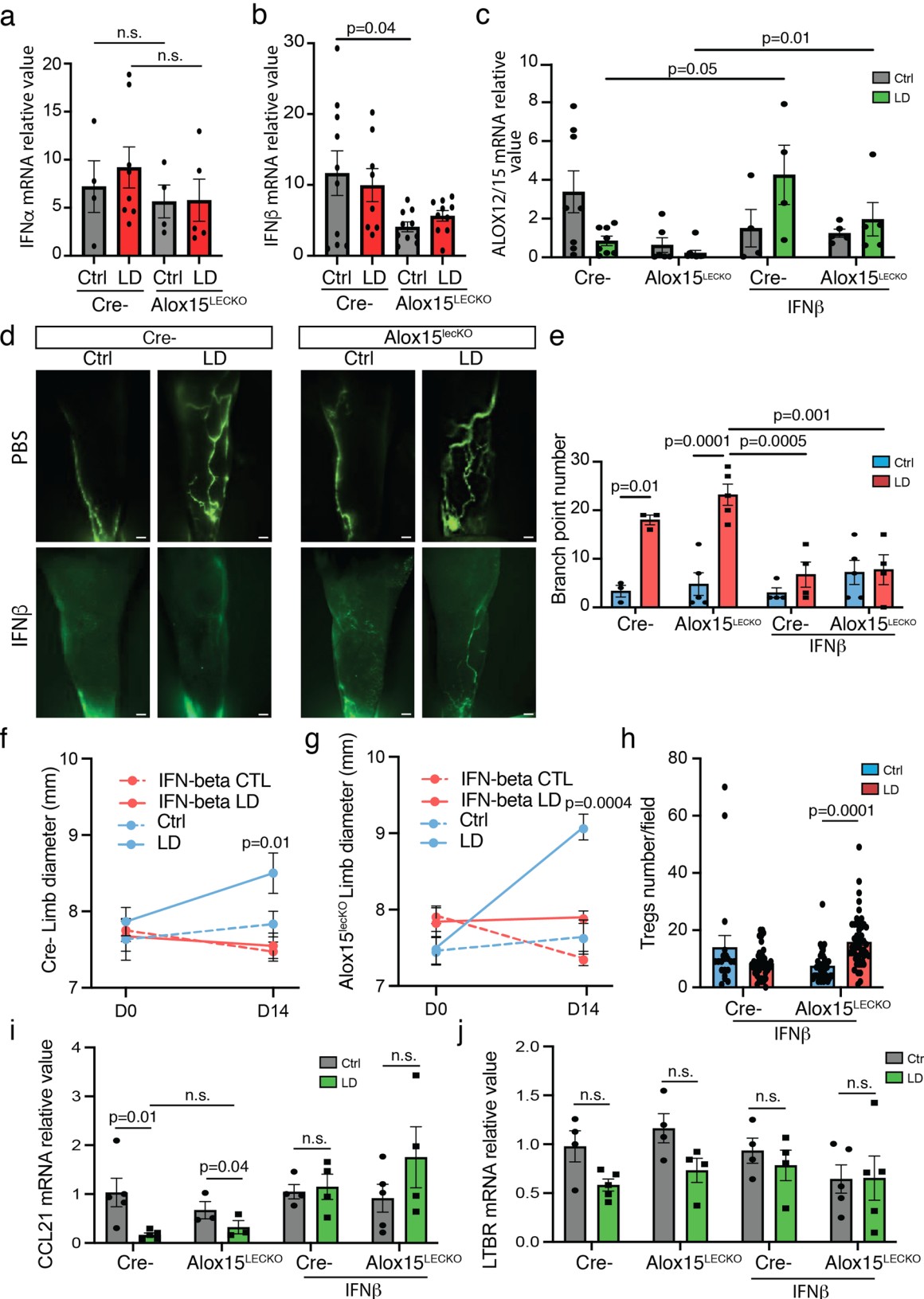

every three days during 10 days starting at the time of LD surgery ($n = 8$). 15-HETE. For LxA4 receptor FPR2 inhibitor, 1 mg/kg/day of WRW4 (Biotechne) was injected IP every 3 days during 10 days in wild-type mice starting at the time of surgery.

**IFN-beta treatment**

ALOX15$^{LECKO}$ mice and control littermates ($n = 4$) were injected IP with 100 ng of recombinant IFNβ (Peprotech) every 2 days during 10 days starting at the time of surgery.

**Fig. 5 | Interferon-beta protects from LD. a** IFN-α mRNA expression in LD skin from ALOX15^LECKO mice. **b** IFN-β mRNA expression in LD skin from ALOX15^LECKO mice. **c** ALOX12/15 gene expression in LD skin from ALOX15^LECKO mice and control littermates (Cre-) treated with IFN-β (100 ng IP, every 2 days for 10 days). **d** Limb lymphography from Cre- (left panel) and ALOX15^LECKO (right panel) mice with LD treated with IFN-β (Scale bar: 1 mm). **e** Quantification of lymphatic branch point in IFN-β-treated ALOX15^LECKO mice and control littermates. **f, g** Quantification of the limb diameter in Cre- (**f**) and ALOX15^LECKO (**g**) mice with LD treated with IFN-β. **h** Quantification of Treg number in LD skin from ALOX15^LECKO mice and control littermates (Cre-) treated with IFN-β. **i** CCL21 mRNA expression in LD skin from ALOX15^LECKO mice treated with IFN-β. **j** LTBR mRNA expression in LD skin from ALOX15^LECKO mice treated with IFN-β. Data are shown as mean ± s.e.m. For **a–n**, $n = 4$–8 mice per group. **a–c, e–j** $p$-values are derived from two-way ANOVA. Source data are provided as a Source data file.

## Histology

Skin samples were frozen immediately in O.C.T. Compound (Cellpath) to be stored at −80 °C until use. Cryosections (5 μm thick) were prepared using CryoSTAP NX50. Frozen tissue sections were fixed in ice-cold acetone for 2 min then left to dry. They were permeabilized with 0.1% triton, and then saturated with 5% BSA in PBS. Samples were incubated with primary antibodies; rat anti-mouse FoxP3 (Abcam Ab20034) 1:100 overnight at 4 °C. Secondary antibodies (Jackson ImmunoResearch 1:500) were incubated for 1 h RT. Apoptotic cells were detected by the use of In Situ Cell Death Detection Kit, Fluorescein (Roche, 11684795910) following the manufacturer's instructions. Finally, DAPI staining was performed and sections were mounted in FMM (DAKO). FoxP3 staining followed the same protocol. For CD31/Lyve-1 staining, primary antibodies were: rat anti-mouse CD31(BD 553370) 1:100 and goat anti-mouse Lyve-1 (Fitzgerald, 70R-LR004) 1:200. All secondary antibodies were purchased from Jackson Immmunoresearch.

TUNEL assay (Roche) was performed according to the manufacturer's instructions.

## Quantitative real-time RT-PCR

Total cellular RNA was isolated from human HDLECs using RNAqueous-Micro Kit (Ambion, USA) according to the manufacturer's instructions. A total of 100 ng RNA was used to synthesize cDNA using the SuperScript VILO cDNA Synthesis Kit (Ambion, USA). The expression of 15-LO, CCL21, SPHK1, SPHK2, S1PR1, LTBR, ICAM1, and VCAM1 was investigated by SYBR Green real-time reverse transcribed polymerase chain reaction using the ABI StepOne+ Real-time PCR System (Applied Biosystems, Villebon s/ Yvette, France). Each reaction was run with 18S as a reference gene and all data were normalized based on the expression levels of 18S. ON-TARGETplus Human ALOX15 siRNA, a guaranteed gene silencing Patented modifications to reduce off-targets, SMARTpool format, and control scramble siRNA were from Dharmacon. Oligonucleotide sequences are provided in Supplementary Table 10.

## Dermis size quantification

Paraffin-embedded tissue sections were colored using Masson's trichrome kit (Biognost) following the manufacturer's instructions. Images were acquired on a Leica microscope using a Leica DFC450C camera. Skin thickness was assessed by analysis using Fiji.

## Transendothelial cell migration assay

Lymphatic transendothelial migration was cultured in the Boyden chamber. Treg cells (500,000) were labeled with CellTracker Orange (CMTMR) according to the manufacturer's directions (Invitrogen) and incubated on top of the confluent monolayer of HDLEC for 2 h. Filters were harvested and cells that passed through the filter were quantified. Statistical significance was determined using Student's t-test.

## Adhesion assay

Adhesion of Treg cells to HDLECs was measured as follows: Treg cells were labeled with CellTracker Orange (CMTMR) according to the manufacturer's directions (Invitrogen) and incubated with monolayers of HDLECs in EBM2-culture medium for 2 h at 37 °C. Cell layers were gently washed with a warmed culture medium and fixed in 3.7% paraformaldehyde prior to the enumeration of bound cells per microscopic field at ×200 magnification. Experiments were performed 4 times; results from representative experiments are shown. Statistical significance was determined using Student's t-test.

**Statistics.** All results presented in this study are representative of at least three independent experiments. Data are shown as the mean ± standard error of the mean (s.e.m.). Statistical significance was determined by two-tailed Student's t-test, two-tailed Mann–Whitney test, or one-way ANOVA with Tukey post hoc test using Prism ver. 6.0 (GraphPad). Differences were considered statistically significant with a $p$-value < 0.05. Data analysis was performed on R Studio version 4.3.1 software.

## Reporting summary

Further information on research design is available in the Nature Portfolio Reporting Summary linked to this article.

## Data availability

The raw reads for scRNA-seq in fastq format have been deposited in the EMBL-EBI database under accession number: MTAB-13019 hosted at Array Express. All other data are available in the article and its supplementary files or from the corresponding author upon request. Source data are provided in this paper.

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

## Acknowledgements

We thank Taija Makinen for providing *Prox1*-CreER^T2 mice. We thank E Lhullier, C. Segura, F. Martins (GenoToul platform), Zakarof A., and Riant E. (TRI platform) for their technical support as well as M. Rousseau from the platform Anexplo Genotoul (Inserm US006, Toulouse, France) for their outstanding technical assistance. We thank the imaging platform of I2MC Institute (R. Flores). We thank Dr. Iacovoni for his input in the bioinformatic analysis of RNAseq data. This work has received funding from the European Union's Horizon 2020 research and innovation program named Theralymph under grant agreement no. 874708. This work has been supported by the Cancéropôle GSO, the Foundation for Medical Research (FRM), the Foundation ARC pour la Recherche contre le Cancer, the National Institute of Cancer (Inca), the Region Midi-Pyrenees FEDER and Midi-Pyrénées REPERE.

## Author contributions

A.Z. performed most of the experiments, analyzed the data, and contributed to experimental design. F.P. contributed to the mouse experimental procedures and immunostaining. E. Benuzzi produced lentivectors. M.N., E. Balzan, L.V. performed revision experiments. F.M., E.L., and A.C.P. contributed to the experimental design. J.G., A.B. contributed to the analysis of A.T. M.D. and V.B. participated in the design of lipidomic experiments. D.N.T. performed the bioinformatic analysis. N.F. contributed to the experimental design of Treg cell

analysis. B.C., J.M.D., and A.B.R. contributed to the human tissue collection and analysis. B.G.S. designed the study, performed data analysis, and wrote the manuscript.

## Competing interests

The authors have submitted a patent application (application numbers EP22305165, EP22305165.7, inventor B. Garmy-Susini) based on the results reported in this study and the authors declare no other competing interests.
