## [Peer Review File · Nature Communications]

15-Lipoxygenase promotes resolution of inflammation in lymphedema by controlling Treg function through IFN- βREVIEWER COMMENTS

Reviewer #1 (Remarks to the Author):

The authors are to be commended for their novel and impactful observations regarding the newly identified role for 15-LO in the resolution of the identified, lymphedema-associated inflammatory signature. The work is considered to be significant to the field, both in its novelty and its potential human therapeutic application. The conclusions are well-supported with the evidence provided and the methodology is sound.

The manuscript is suitable for publication, but its impact might be enhanced by an expanded discussion to elucidate the relationship of these findings to other, published experimental approaches related to the pathogenesis and attempted therapeutics of the tissue pathology associated with lymphedema. It would also be useful to emphasize the potential, albeit theoretical drawbacks and barriers associated with targeting 15-LO as a human therapeutic target.

Reviewer #2 (Remarks to the Author):

The manuscript by Zamora et al. describes a novel role for pro-resolving pathways in limiting lymphedema (LD) in humans and mice. The study shows important differences in 15-LO properties and enzymatic activity during LD that limit pro-resolving lipid mediator production. This in turn limits Treg numbers in adipose tissues, which is critical for the restoration of normal lymphatic function. The study is very novel and interesting with a new mouse strain of conditional 12/15-LO KO in LECs. However, the limited scope of the study and the overreaching conclusions drawn by the authors prevent publication at this stage. This can be changed if the following issues are rectified.

Major comments:

1. In Fig. 2 the authors show a reduction in the pro-inflammatory LM LTB₄ in LD tissues. Does that lead to a reduction in PMN numbers. Also how do they explain the increase in CCL2 is not associated with an increase in monocyte/macrophage numbers?
2. In Fig. 2 there is a clear reduction in the molecular weight of 15-LO which might account for the differences in its products in LD tissues. What is the nature of this reduced molecular

weight? Can this modification account for the reduced activity?

3. In Fig. 3d the percentage of CD4 T cells seem to reduce in LD samples while the data in ext Fig. 4 indicates the opposite. Please clarify. Also, the gate used for Tregs in this panel is of different size. Please correct the gating and the corresponding data.

4. The authors suggest that SPM production by 15-LO is key in limiting LD but do not provide direct evidence to support this hypothesis or for a reduction in SPM production in LEC KOs. Some of the SPM, like LXA4 or RvD1 can be obtained and tested for their actions in the mouse model to identify the lipid mediators involved in the resolution of LD and whether they can rescue the exacerbated outcomes in 12/15-LO cKOs. Alternatively, antagonists of the corresponding receptors can be examined for their ability to exacerbate disease. This is particularly important in deciphering the role of LTB4 and its receptors.

5. It was recently shown that IFN- β , produced by Pro resolving macrophages induces IL-10, and 12/15-LO expression in dermal cells and that parallels with the rescue of a hyperfibrotic phenotype and hair loss in ACKR2-deficient mice. Type I IFNs are also well-established enhancers of Treg differentiation and function. Are type I IFNs also deficient in LD and can this disorder be treated by recombinant IFN- β in a 12/15-LO-dependent manner. This issue should be tested experimentally and discussed in the discussion.

6. Since the migration of Tregs and the function of PPAR γ were not tested in the manuscript (see minor comments) I suggest changing its title to “15-Lipoxygenase drives inflammation resolution in lymphedema by controlling Treg function”

Minor comments: 1. Line 55- change to possesses a distinct phenotype and specifically express the transcription factor PPAR- γ .

2. Line 99- change to – and IL-6, a major pro-inflammatory cytokine.

3. In the legend of Ext data Table 1- ? should be replaced with .

4. Line 114- change to responsible for.

5. Lines 122, 143 - Change to pro-resolving.

6. In Fig 1g some genes appear twice (SAMS1). Please explain.

7. Line 126- delete expression.

8. The authors indicate they did not find any difference in SPMs generated by DHA metabolism. This statement should be replaced with “the DHA metabolites showed inverse differences with RvD1 reduced and RvD2 increased in LD tissues”.

9. The author interchange the terms ALOX15, 15 LOX and 15-LO. Please make sure the right

terms are used consistently for genes and proteins.

10. Fig. 2h shows expression of 15-LO in LD endothelial cells but did not show their healthy controls (which should be increased). Please provide this image as well.

11. In ext Fig. 3g the legend text is missing. In panel j the overlap between Lyve1 and 12/15-LO (the correct name for the mouse protein, please correct in the text and figure) is very low. The statement in line 144 should be corrected. The lipid panels are very hard to evaluate. Please add numbers to the color diagrams and detail the data points in panels f-g.

12. Line 635- correct to 17-HDOHE.

13. In ext Fig. 4b the panels should be labeled properly. Quantification of collagen content, reduced S.C. fat tissue thickness and number of hair follicles would strengthen the enhanced fibrosis conclusion (the same goes for Fig. 4d).

14. Line 169- at the end of the sentence add "even in the absence of an inflammatory process". Also, since the difference in Treg numbers in the KO seems to be due to apoptosis rather than infiltration, please change recruitment on line 146, and infiltration on lines 152 and 158 to "numbers". In the legend title for Fig. 3 change to T cell function

15. Line 175- please change to lentivector rescues Treg numbers...

16. Line 415- correct to (FoxP3).

17. A quantification of 12/15-LO (protein) in LECs should be added to both the conditional-KO and the lentivector data.

18. Line 186- change whereas to whether.

19. Line 189- change chemoattractive cytokines to chemokine.

20. Line 194- change overexpression to upregulation.

21. In figure 4 panels f and g should both quantify cell numbers. Panels h-j are not relevant to the manuscript and should be replaced with equivalent data from 12/15-LO knockout mice.

22. Line 202- delete "it".

23. Line 206- change obvious to conceivable and delete during.

24. Line 208- change candidates to treat to treatment to.

25. The sentence "Here, we found that 15-HETE induces CCL21 synthesis by lymphatic endothelial cells to stimulate Treg chemoattraction" and the following sentence are inaccurate and overreaching. Please correct.

26. In figure 2h, quantification of colocalization should be added.
27. Line 448- change to paired.
28. Line 475- change form to from.
29. Line 488- please insert citation of the reference.
30. Line 618- replace ? with .

Reviewer #3 (Remarks to the Author):

Lymphedema (LD) is characterized by accumulation of lipids and inflammatory cell infiltrates in the limb. In this study, the authors investigated how lipid mediators are modulated during LD and how they may impact resolution of inflammation by affecting regulatory T cells (Tregs). They found that LD tissues showed decrease in arachidonic acid-derived lipid mediators generated by the 15-lipoxygenase (15-LO) and a reduction in Tregs. Ablation of ALOX15 in the lymphatic system led to aggravation of LD, which can be rescued by injection of ALXO15-expressing lentivectors. While there are some interesting phenotypic observations on the change of lipid mediators during LD, this study lacks sufficient novel mechanistic insights in how these lipid mediators and ALXO15 impact LD. The involvement of PPARg+ Tregs was emphasized throughout the manuscript, but this study lacks direct evidence that modulation of these Tregs was the major mechanism by which these lipid mediators suppress LD. It was also not clear how these lipid mediators impacted PPARg+ Tregs specifically. In addition, there are several technical issues. Below are the specific points:

Major points:

1. Tregs are reduced in LD limb or with ALOX15 conditional KO. However, is this the main reason that these mice develop more severe LD? The authors need to use more specific Treg targeting strategies (eg, Foxp3-DTR mice, IL-2/anti-IL-2 complex injection, adoptive Treg transfer...) in these mouse models to show that the modulation on Tregs is causal for the LD phenotype observed.
2. While PPARg+ Tregs were known to reside in the visceral adipose tissue of mice and control systemic metabolic processes such as insulin sensitivity, their role in controlling resolution of LD was not characterized before. Therefore, the authors need to perform gain-

and loss-of-function studies to show whether and how this specific subset of Tregs regulate LD.

3. It was also not clear how LD or ALXO15 affect Tregs. While in figure 3 cell apoptosis was emphasized (line 171), in figure 4 it was attributed to differences in cell recruitment (line 184). How would modulating ALXO15 in lymphatics impact apoptosis of Tregs? Based on the introduction and literatures, PPARg+ Tregs were considered adipose-tissue resident. Were these cells already present in the tissue or were they recruited to the tissue following LD? If the later, by what chemokines? Also, does LD or ALXO15 KO affect proliferation of these Tregs?

4. Why are PPARg+ Tregs specifically but not overall Tregs impacted by 15-LO expressing lentivector injection?

5. Are any of the lipid mediators modulated during LD natural ligands for PPARg? If so, how will that impact the PPARg+ Tregs and other PPARg expressing cells (eg, macrophages)?

6. The manuscript lacks quality controls for many of the flow cytometry data. How are each immune population gated? How were PPARg+ Tregs identified? Commercially available PPARg antibodies were not known to work well with flow cytometry. The authors need to show representative plot of PPARg staining in Tregs.

Minor points:

1. In Fig 1C, the volcano plots without highlighting any specific genes for each individual patients will not tell whether similar genes were modulated in different individuals. A PCA analysis here will be helpful to show the overall similarities in transcriptome between samples.

2. Fig 1J and K, please show gating strategy and representative flow plot of how each immune subsets are identified. Are cell number normalized by any parameters, such as weight of tissue? In addition, based on figure 1K, many cell types are increased, not just CD4+ T cells. In the text (line 110-111), overall T cells should be defined as CD3+, not CD4+.

3. Extended figure 2. The change of lipids in adipose tissue is not very obvious. Is this statistically significant? It is difficult to evaluate this figure without proper statistical analysis.

4. Fig 2e. Why is the protein size of ALOX15 different in Ctl and LD samples? Are there any posttranslational modifications of this protein?

5. Extended figure 3f and g. There are no annotations of what each color in the graph means. Also, 17 HDOHE seems to also be reduced like 15 HETE. Is it significant?
6. Extended figure 4a. Difference in limb diameter is marginal.
7. Extended figure 4f. The flow plot is very hard to see. No numbers were shown on the gate to indicate frequencies of cells within the gate.
8. Extended figure 4g, not sure what the authors mean by CD4+ cells (%CD4+). % CD4+ T cells among what cells?
9. Figure 3d. Again, no number of the gates to indicate frequency of cells in each gate. Is the frequency/percentage of Tregs also reduced in LD? What about changes in other immune cells (Tconvs, CD8+ T cells, macrophages, DCs?).
10. No data showing efficiency and specificity of Alox15 KO following tamoxifen injection.
11. Figure 3h, some dots were shifted to the right of the plot, maybe during figure editing.
12. Figure 3i. Percent of foxp3+PPARg+ cells among what cells?
13. Figure 4j, CCL21 and LTBR are reduced upon siALOX15 knockdown in primary cultures of HDLEC. Are these changes validated in ALOX15 conditional KO mice? More importantly, is this responsible for the reduction in Treg recruitment?

Reviewer #1 (Remarks to the Author):

The authors are to be commended for their novel and impactful observations regarding the newly identified role for 15-LO in the resolution of the identified, lymphedema-associated inflammatory signature. The work is considered to be significant to the field, both in its novelty and its potential human therapeutic application. The conclusions are well-supported with the evidence provided and the methodology is sound.

The manuscript is suitable for publication, but its impact might be enhanced by an expanded discussion to elucidate the relationship of these findings to other, published experimental approaches related to the pathogenesis and attempted therapeutics of the tissue pathology associated with lymphedema. It would also be useful to emphasize the potential, albeit theoretical drawbacks and barriers associated with targeting 15-LO as a human therapeutic target.

We thank reviewer 1 for supporting our study and emphasizing the crucial role of the lymphatic endothelial 15-LO in the resolution of inflammation.

According to reviewer 1 instructions, discussion was improved: "However, targeting 15-LO as a human therapeutic target may have certain barriers and challenges associated with it. In particular, due to the complexity of the arachidonic acid metabolic pathway involving multiple enzymes and metabolites that may have unintended consequences on other enzymes. Also, ensuring effective delivery of 15-LO-derived SPM to the specific tissues or cells of interest can be challenging to exert the desired therapeutic effect while minimizing systemic distribution. Despite these barriers, targeting 15-LO remains an area of active research, and ongoing studies continue to explore its therapeutic potential in various diseases. Here, we found that 15-HETE reduces LD. Importantly, using ALOX15^{LECKO} mice, we have identified the crucial role of the lymphatic endothelial production of 15-HETE as a key regulator in lymphedema. No significant beneficial effect was observed with RvD1 and LxA4. However, we treated the mice systemically. It would be interesting to continue these studies with local treatments at the LD limb level in order to draw conclusions on effectiveness. In parallel, we identified that IFN β , a recently identified pro-resolving cytokine in skin fibrosis³¹, which may represent an exciting alternative for LD resolution treatment. These molecules, alone or in combination, could also constitute good candidates for cardiovascular pathologies in which the lymphatic system is damaged or dysfunctional, such as cardiac ischemia, atherosclerosis, or chronic inflammatory diseases⁴¹⁻⁴³"

Reviewer #2 (Remarks to the Author):

The manuscript by Zamora et al. describes a novel role for pro-resolving pathways in limiting lymphedema (LD) in humans and mice. The study shows important differences in 15-LO properties and enzymatic activity during LD that limit pro-resolving lipid mediator production. This in turn limits Treg numbers in adipose tissues, which is critical for the restoration of normal lymphatic function. The study is very novel and interesting with a new mouse strain of conditional 12/15-LO KO in LECs. However, the limited scope of the study and the overreaching conclusions drawn by the authors prevent publication at this stage. This can be changed if the following issues are rectified.

We thank Reviewer 2 for the careful evaluation of the manuscript, we have now completed the study with experiments in order to answer to all the questions raised.

Major comments:

1. In Fig. 2 the authors show a reduction in the pro-inflammatory LM LTB₄ in LD tissues. Does that lead to a reduction in PMN numbers. Also how do they explain the increase in CCL2 is not associated with an increase in monocyte/macrophage numbers?

Additional flow cytometry analysis of human tissue biopsies was performed. It revealed no difference in macrophages number (CD206+ and CD206-) and in PMN neutrophil number (now in Extended Fig. 1f). We agree with reviewer 2, the lack of regulation of macrophages number was surprising (FACS

gating strategy is now provided in Figure 1c-e). However, the study compared uninjured arm to lymphedema arm in the same patient who developed LD more than 5 years ago and probably have chronic systemic inflammation. We would expect to observe a difference in patients who developed the pathology compared to patients without lymphedema.

2. In Fig. 2 there is a clear reduction in the molecular weight of 15-LO which might account for the differences in its products in LD tissues. What is the nature of this reduced molecular weight? Can this modification account for the reduced activity?

Although the primary structure contains a number of potential phosphorylation sites there is no evidence that protein phosphorylation/dephosphorylation constitutes a regulatory element of cellular ALOX15 activity (Igor Ivanov 2015 PMID: 26216303). Also, studies have shown that 15-LO glycosylation is not required for its catalytic activity (Ivanov I. Gene 2015). However, lipoxygenase needs changes from ferrous to ferric species to be activated (Wiesner R. FEBS Lett. 1996). Ferrous 15-LO is oxidized by Nitric Oxide (NO) to a pre-activated ferric form thus changing the isoelectric point that determines the electrophoretic mobility in the gel. This is the main hypothesis to explain the change in molecular weight observed in Fig 2.

This comment was added to the result and discussion sections of the manuscript.

3. In Fig. 3d the percentage of CD4 T cells seem to reduce in LD samples while the data in ext Fig. 4 indicates the opposite. Please clarify. Also, the gate used for Tregs in this panel is of different size. Please correct the gating and the corresponding data.

The difference observed in CD4+ cells signal is attributed to the total amount of cell count. We only analyzed the upper limb adipose tissue from mice, which corresponds to 30-40 mg of tissue. When normalized to the tissue weight, we observed an increase of CD4+ T cell population in LD whereas Tregs were downregulated.

Gating was corrected in Figure 3.

4. The authors suggest that SPM production by 15-LO is key in limiting LD but do not provide direct evidence to support this hypothesis or for a reduction in SPM production in LEC KOs. Some of the SPM, like LXA4 or RvD1 can be obtained and tested for their actions in the mouse model to identify the lipid mediators involved in the resolution of LD and whether they can rescue the exacerbated outcomes in 12/15-LO cKOs. Alternatively, antagonists of the corresponding receptors can be examined for their ability to exacerbate disease. This is particularly important in deciphering the role of LTB4 and its receptors.

To investigate which SPM could mediate 15-LO effect on LD, mice received IP injections Resolvin D1 (RvD1), lipoxin A4 (LxA4), or their receptor inhibitor WRW4 (Fig. 4i, Extended Data Fig. 7e). We did not observe any effect of RvD1, LxA4 and WRW4 on mice swelling (Fig. 4i, Extended Data Fig. 7e).

In contrast, 15HETE treatment completely inhibited LD in ALOX15^{LECKO} mice and in control littermates (Fig. 4i) suggesting that 15HETE mediates the pro-resolving effect of 15-LO in the lymphatic endothelium.

5. It was recently shown that IFN- β , produced by Pro resolving macrophages induces IL-10, and 12/15-LO expression in dermal cells and that parallels with the rescue of a hyperfibrotic phenotype and hair loss in ACKR2-deficient mice. Type I IFNs are also well-established enhancers of Treg differentiation and function.

Are type I IFNs also deficient in LD and can this disorder be treated by recombinant IFN- β in a 12/15-LO-dependent manner. This issue should be tested experimentally and discussed in the discussion.

According to reviewer 2 instructions, we investigated Type I IFN expression in ALOX15^{LECKO} mice. The knock down of ALOX15 had no effect on IFN α expression, but reduced IFN β expression in the skin and adipose tissue (Extended data Fig. 6a and b). Interestingly, the expression was not affected in LD. These

results could explain the increase of LD observed in ALOX15^{LECKO} mice compared to control Cre-littermates.

We next investigated the effect of recombinant IFN β in ALOX15^{LECKO} mice (Extended data Fig. 8a and b). Mice were injected 3 times per week by 100ng of rIFN β during 10 days. IFN β treatment significantly reduced limb swelling in ALOX15^{LECKO} mice and control mice (Extended data Fig. 8c and d). This was associated with a strong decrease in dermal backflow (Extended data Fig. 8e and f).

Importantly, Treg cell numbers were increased in LD limb (Extended data Fig. 8g).

These results have been added to the manuscript and discussed in the discussion.

6. Since the migration of Tregs and the function of PPAR γ were not tested in the manuscript (see minor comments) I suggest changing its title to “15-Lipoxygenase drives inflammation resolution in lymphedema by controlling Treg function”

We agree with reviewer 2, title has been modified accordingly.

However, some more experiments were added to the manuscript to improve our hypothesis concerning PPAR γ positive Tregs. Flow cytometry analysis of Tregs from adipose depots revealed that 75.9% of limb Tregs are PPAR γ +, and decreased to 66% in LD (Data provided in Extended Fig 7b).

Minor comments:

1. Line 55- change to possesses a distinct phenotype and specifically express the transcription factor PPAR- γ .

Modification was performed

2. Line 99- change to – and IL-6, a major pro-inflammatory cytokine.

Modification was performed

3. In the legend of Ext data Table 1- ? should be replaced with.

Additional information is now provided in the legend: “normal arm vs lymphedema arm”

4. Line 114- change to responsible for.

Modification was performed

5. Lines 122, 143 - Change to pro-resolving.

Modification was performed

6. In Fig 1g some genes appear twice (SAMSN1). Please explain.

SAMSN1 appears twice in the table, as there are several variants of this gene. Referring to the accession numbers (NM_001256370) and (NM_001286523, NM_022136), there are three variants produced from two alternative promoters (see screen capture). The first variant (NM_001256370 called variant 2) is made up of 9 exons from an upstream promoter, while the other two variants (NM_001286523, NM_022136 called variant 1 and variant 3 respectively) are made up of 8 or 9 exons but transcribed from a downstream promoter. The amount of transcript produced from these two promoters is likely to be slightly different, requiring two separate lines in the table. We propose to indicate v2 and v1+v3 along with the gene name to clarify this in the table.

7. Line 126- delete expression.

Modification was performed

8. The authors indicate they did not find any difference in SPMs generated by DHA metabolism. This statement should be replaced with “the DHA metabolites showed inverse differences with RvD1 reduced and RvD2 increased in LD tissues”.

Modification was performed

9. The author interchange the terms ALOX15, 15 LOX and 15-LO. Please make sure the right terms are used consistently for genes and proteins.

Modification was performed

10. Fig. 2h shows expression of 15-LO in LD endothelial cells but did not show their healthy controls (which should be increased). Please provide this image as well.

Images are now provided in Fig. 2h

11. In ext Fig. 3g the legend text is missing. In panel j the overlap between Lyve1 and 12/15-LO (the correct name for the mouse protein, please correct in the text and figure) is very low. The statement in line 144 should be corrected. The lipid panels are very hard to evaluate. Please add numbers to the color diagrams and detail the data points in panels f-g.

We apologize for the missing legend in Fig 3e, it has been corrected. Line 144 was corrected; numbers were provided on heatmap and data points are now provided in panels f-g.

12. Line 635- correct to 17-HDOHE.

Correction was performed.

13. In ext Fig. 4b the panels should be labeled properly. Quantification of collagen content, reduced S.C. fat tissue thickness and number of hair follicles would strengthen the enhanced fibrosis conclusion (the same goes for Fig. 4d).

Figure is now correctly labelled. Discussion was improved relating the fibrosis conclusion and IFNbeta experiments.

14. Line 169- at the end of the sentence add “even in the absence of an inflammatory process”. Also, since the difference in Treg numbers in the KO seems to be due to apoptosis rather than infiltration, please change recruitment on line 146, and infiltration on lines 152 and 158 to “numbers”. In the legend title for Fig. 3 change to T cell function

Corrections have been performed on the manuscript.

15. Line 175- please change to lentivector rescues Treg numbers...

Correction was performed.

16. Line 415- correct to (FoxP3).

Correction was performed.

17. A quantification of 12/15-LO (protein) in LECs should be added to both the conditional-KO and the lentivector data.

Quantification was added to figure 4 (lentivector) and extended figure 5 (conditional-KO)

18. Line 186- change whereas to whether.

Correction was performed.

19. Line 189- change chemoattractive cytokines to chemokine.

Correction was performed.

20. Line 194- change overexpression to upregulation.

Correction was performed.

21. In figure 4 panels f and g should both quantify cell numbers. Panels h-j are not relevant to the manuscript and should be replaced with equivalent data from 12/15-LO knockout mice.

The effect of ALOX15-overexpressing lentivector was quantified in 12/15-LO knockout mice. We observed a strong reduction of dermis thickness as previously observed in WT mice.

22. Line 202- delete “it”.

Correction was performed.

23. Line 206- change obvious to conceivable and delete during.

Correction was performed.

24. Line 208- change candidates to treat to treatment to.

Correction was performed.

25. The sentence “Here, we found that 15-HETE induces CCL21 synthesis by lymphatic endothelial cells

to stimulate Treg chemoattraction” and the following sentence are inaccurate and overreaching. Please correct.

Based on the new data provided, we changed the sentence for “Here, we found that 15-HETE reduces LD.”

26. In figure 2h, quantification of colocalization should be added.

The quantification is provided in figure 2g.

27. Line 448- change to paired.

Corrected

28. Line 475- change form to from.

Corrected

29. Line 488- please insert citation of the reference.

Reference was added to the manuscript

30. Line 618- replace ? with .

Corrected by: between conditions (normal arm vs lymphedema arm).

Reviewer #3 (Remarks to the Author):

Lymphedema (LD) is characterized by accumulation of lipids and inflammatory cell infiltrates in the limb. In this study, the authors investigated how lipid mediators are modulated during LD and how they may impact resolution of inflammation by affecting regulatory T cells (Tregs). They found that LD tissues showed decrease in arachidonic acid-derived lipid mediators generated by the 15-lipoxygenase (15-LO) and a reduction in Tregs. Ablation of ALOX15 in the lymphatic system led to aggravation of LD, which can be rescued by injection of ALXO15-expressing lentivectors. While there are some interesting phenotypic observations on the change of lipid mediators during LD, this study lacks sufficient novel mechanistic insights in how these lipid mediators and ALXO15 impact LD. The involvement of PPARg+ Tregs was emphasized throughout the manuscript, but this study lacks direct evidence that modulation of these Tregs was the major mechanism by which these lipid mediators suppress LD. It was also not clear how these lipid mediators impacted PPARg+ Tregs specifically. In addition, there are several technical issues. Below are the specific points:

We thank reviewer 3 for the careful evaluation of the manuscript. We agree with the way we emphasized the importance of PPARg+ Treg cells, whereas the originality of this manuscript should be focused on the discovery of the role of inflammation resolution in lymphedema. Therefore, even if we reinforce our conclusion about the subpopulation of T regulatory cells in LD by additional experiments, we decided to change the title accordingly with reviewer 2 recommendation. The title of the manuscript is now: “15-Lipoxygenase drives inflammation resolution in lymphedema by controlling Treg function”

Major points:

1. Tregs are reduced in LD limb or with ALOX15 conditional KO. However, is this the main reason that these mice develop more severe LD? The authors need to use more specific Treg targeting strategies (eg, Foxp3-DTR mice, IL-2/anti-IL-2 complex injection, adoptive Treg transfer...) in these mouse models to show that the modulation on Tregs is causal for the LD phenotype observed.

To confirm the role of Treg cells in LD, we performed Treg adoptive transfer using Treg from DEREK mice.

Briefly, GFP+ Treg cells were isolated from the spleen of DEREK mice. 10E3 cells were injected in the adipose tissue limb from control (no surgery) and lymphedema limb from Cre- and ALOX15^{LECKO} mice (Extended data figure 6i and j). We observed a significant decrease of lymphedema in both groups even if the transplanted cells were isolated from the spleen. Additional analysis of the limb adipose

tissue with an anti-PPAR γ antibody revealed that 75.9% of Treg cells in AT are PPAR γ +, and the number decreases to 66% in LD. This was added in Extended data Fig. 7b and in the discussion of the manuscript.

2. While PPAR γ + Tregs were known to reside in the visceral adipose tissue of mice and control systemic metabolic processes such as insulin sensitivity, their role in controlling resolution of LD was not characterized before. Therefore, the authors need to perform gain- and loss-of-function studies to show whether and how this specific subset of Tregs regulate LD.

We had unfortunately no possibility to perform adoptive transfer of PPAR γ + Treg cells as PPAR γ is a nuclear protein and does not allow to perform cell sorting. However, to emphasize the role of PPAR γ + Treg cells, we performed flow cytometry analysis of limb adipose tissue from DERE γ mice that undergo lymphedema surgery. We found that 75.9% of Tregs are PPAR γ + in limb adipose tissue (Extended data Fig. 7b).

Then, we performed adoptive Treg transfer from DERE γ mice. We confirmed that this Treg subpopulation number is decreased in LD due to an increase of apoptosis (Figure 3k-m, Extended data Fig. 7a).

3. It was also not clear how LD or ALXO15 affect Tregs. While in figure 3 cell apoptosis was emphasized (line 171), in figure 4 it was attributed to differences in cell recruitment (line 184). How would modulating ALXO15 in lymphatics impact apoptosis of Tregs?

We agree with reviewer 3. First, we over-concluded to an increase of cell recruitment, whereas we did not prove this statement. Therefore, we changed the paragraph title from “rescue Treg recruitment” for “rescue Treg numbers” according to reviewer 2 instructions.

Then Figure 3K shows an increase in apoptotic Treg in ALOX15^{LECKO} mice compared to control littermates showing that lymphatic endothelial 15-LO directly impacts Treg survival.

To understand whether ALOX15 in lymphatics may impact apoptosis of Tregs, we performed in vitro apoptosis experiment on sorted Treg cells incubated with 15-HETE, the 15-LO generated SPM. We found that 15-HETE significantly protects Tregs from apoptosis (Extended data Figure 7h), but had no effect on Treg cells adhesion to the lymphatic monolayer or trans-lymphatic endothelial migration (Extended data Figure 7f and g).

This is in line with in vivo new experiments provided in Figure 3i: ALOX15^{LECKO} mice were treated with 15-LO SPM RvD1, LxA4 and 15HETE. We observed a significant reduction of LD with 15HETE, whereas resolving D1 and LxA4 had no effect suggesting a selective effect of 15-HETE on the Treg cells survival in lymphedema.

Based on the introduction and literatures, PPAR γ + Tregs were considered adipose-tissue resident. Were these cells already present in the tissue or were they recruited to the tissue following LD?

Additional experiment was performed in DERE γ mice: flow cytometry analysis of PPAR γ positive Tregs was analyzed in limb adipose depots and compared to subcutaneous inguinal adipose depot. We found that 75.9% of Tregs are PPAR γ + in limb adipose tissue suggesting that these Tregs are resident.

In parallel, we quantified PPAR γ + Treg cells in limb AT on histological sections. We found that the number of PPAR γ + Treg cells decrease during LD and/or in ALOX15^{LECKO} mice limb AT (Figure 3m). These data show that PPAR γ + Treg cells are present in normal limb AT.

If the later, by what chemokines?

It was recently shown that IFN- β , produced by Pro resolving macrophages induces IL-10, and 12/15-LO expression in dermal cells and that parallels with the rescue of a hyperfibrotic phenotype and hair loss in ACKR2-deficient mice. Type I IFNs are also well-established enhancers of Treg differentiation and function.

We investigated Type I IFN expression in ALOX15^{LECKO} mice. The knock-down of ALOX15 had no effect on IFNalpha, but significantly reduced IFNbeta expression (Extended data Fig.8a and b).

Interestingly, the expression was not affected by LD. These results could explain the increase of LD observed in ALOX15^{LECKO} mice compared to control Cre- littermates.

We next investigated the effect of recombinant IFNbeta in ALOX15^{LECKO} mice (Extended data Fig. 8c and d). Mice were injected 3 times per week with 100ng of recombinant IFNbeta during 10 days. IFNbeta treatment significantly reduced limb swelling in ALOX15^{LECKO} mice and control mice (Extended data Fig. 8c and d). This was associated with a strong decrease in dermal backflow (Extended data Fig. 8 e and f).

Also, we found a downregulation of CCL21, the lymphatic endothelial cytokine.

Also, does LD or ALXO15 KO affect proliferation of these Tregs?

To investigate if 15-LO could control Treg cells proliferation, we performed immunodetection of the proliferative marker KI67 on ALOX15^{LECKO} mice.

We found that despite a lower number of Treg cells in LD tissues, LD had no effect of Treg cell proliferation (Extended data Fig. 7c and d). Also, the knock-down of 15-LO in LEC did not affect Treg cells proliferation suggesting that protective effect of 15-LO derivative was restricted to an effect on survival, but not proliferation (Extended data Fig. 7c and d).

4. Why are PPARg+ Tregs specifically but not overall Tregs impacted by 15-LO expressing lentivector injection?

The overall message of the current manuscript aims at demonstrating that Treg cell population plays a crucial role on LD, with a greater importance in PPARg+ subpopulation as we observed decrease in both FoxP3+ and FoxP3-PPARg+ cells in LD. We explored FoxP3-PPARg+ cells due to the etiology of LD that consists in an accumulation of adipose tissue in the limb. However, we can observe an effect on total Treg cell population. For this reason, the title of the manuscript was changed according to reviewer 2 for a more general message: "15-Lipoxygenase drives inflammation resolution in lymphedema by controlling Treg function".

However, there is a direct effect of 15-LO on PPARg positive cells as 15-LO derived SPM 15-HETE was reported to bind and activate PPARγ in both human and murine macrophages (Huang et al., 1999) (Ryan G. Snodgrass*Frontiers Pharmacol 2019) as well as lymphocytes (Je-Min Choi Mol Cells. 2012).

5. Are any of the lipid mediators modulated during LD natural ligands for PPARg? If so, how will that impact the PPARg+ Tregs and other PPARg expressing cells (eg, macrophages)?

Among the 15-LO derived lipid mediators, LTB4 is known to bind PPARalpha (Elena Rigamonti ATVB 2008), whereas RvD1 and 15-HETE both bind PPARgamma (Haifa Xia Biomed Res Int 2019 ; Ryan G. Snodgrass* Frontiers Pharmacol 2019).

ALOX15^{LECKO} mice were treated with SPM. We found that LTB4 and RvD1 had no effect on LD, whereas we observed a beneficial effect of 15-HETE (New data provided in Fig4i) suggesting that 15-HETE plays a crucial role in the current process.

Also, 15-HETE is known to increase the expression of PPAR-gamma messenger RNA (mRNA) in many cell types including vascular smooth muscle cells (Limor R 2008), CHO cells (significantly weaker extent Simone Naruhn Mol Pharmacol 2010). More importantly, 15-HETE was reported to bind and activate PPARγ in both human and murine macrophages (Huang et al., 1999) (Ryan G. Snodgrass*Frontiers Pharmacol 2019) as well as lymphocytes (Je-Min Choi Mol Cells. 2012).

Flow cytometry analysis was performed on dermolipectomy tissue samples. We do not observe any difference in macrophages count in lymphedema arm compared to the normal arm (Fig. 1i). Additional data were provided in Extended Figure 1c-e to provide the gating strategy. In parallel, additional flow

cytometry analysis of human tissue biopsies revealed no difference in macrophages number (CD206+ and CD206) and in PMN neutrophil number (Extended Fig. 1f) suggesting that macrophages are not the major target of 15-HETE in lymphedema.

To determine whether 15-HETE could impact Treg cells in LD, we performed in vitro co-culture experiments between LEC and Treg cells sorted from DERE mice. We found that 15-HETE had no effect on Treg cells transendothelial lymphatic migration and adhesion (Extended data Figure 7f and g). However, 15-HETE significantly inhibited Treg cell apoptosis (Extended data Figure 7h).

6. The manuscript lacks quality controls for many of the flow cytometry data. How are each immune population gated? How were PPARg+ Tregs identified? Commercially available PPARg antibodies were not known to work well with flow cytometry. The authors need to show representative plot of PPARg staining in Tregs.

We apologize for the lack information concerning the gating strategy of immune population, it has been correctly and gating strategy is now added to the method and to the Figure.

PPARg+ Treg identified by histology, representative panel images were added to the manuscript in Extended Figure 7a. In parallel, we performed additional flow cytometry analysis of adipose tissue from DERE mice using PPARg antibody. Data are now provided in Extended data Fig.7b.

Minor points:

1. In Fig 1C, the volcano plots without highlighting any specific genes for each individual patients will not tell whether similar genes were modulated in different individuals. A PCA analysis here will be helpful to show the overall similarities in transcriptome between samples.

PCA analysis is now provided in Extended data Figure 1a. Upregulated and downregulated genes are provided in Figure 1 g-j and in Extended data Fig. 1b.

All the RNA-seq raw data is available in ArrayExpress using the link <https://www.ebi.ac.uk/biostudies/arrayexpress/studies/E-MTAB-13019?accession=E-MTAB-13019>

2. Fig 1j and K, please show gating strategy and representative flow plot of how each immune subsets are identified. Are cell number normalized by any parameters, such as weight of tissue? In addition, based on figure 1K, many cell types are increased, not just CD4+ T cells. In the text (line 110-111), overall T cells should be defined as CD3+, not CD4+.

Gating strategy and representative flow plots are now provided in Extended data Figure 1c-e and in the material and methods of the manuscript.

Cells are expressed as number/g adipose tissue. It is now clearly mentioned in the legend of the Y axis of Figure 1j and k.

Based on two-way Anova followed by Sidak multiple comparison test, the CD3+ T and the CD4+ T cell numbers are statistically significant higher in patient with lymphedema.

3. Extended figure 2. The change of lipids in adipose tissue is not very obvious. Is this statistically significant? It is difficult to evaluate this figure without proper statistical analysis.

The amount of SPM in adipose tissue was not significantly reduced in LD (Extended Figure 2) compared to the SPM quantified in the LD skin (Figure 1). P value are now added in Figure 1 and Extended Figure 2 legend

4. Fig 2e. Why is the protein size of ALOX15 different in Ctl and LD samples? Are there any posttranslational modifications of this protein?

Although the primary structure contains a number of potential phosphorylation sites there is no evidence that protein phosphorylation/dephosphorylation constitutes a regulatory element of cellular ALOX15 activity (Igor Ivanov 2015 PMID: 26216303). Also, studies have shown that 15-LO glycosylation is not required for its catalytic activity (Ivanov I. Gene 2015).

However, lipoxygenase needs changes from ferrous to ferric species to be activated (Wiesner R. FEBS Lett. 1996). Ferrous 15LO is oxidized by Nitric Oxide (NO) to a pre-activated ferric form thus changing the isoelectric point that determines the electrophoretic mobility.

This part was added to the discussion of the manuscript.

5. Extended figure 3f and g. There are no annotations of what each color in the graph means. Also, 17 HDOHE seems to also be reduced like 15 HETE. Is it significant?

We apologize for the missing legend, it is now corrected. We agree with reviewer 3, we observe a decrease of 17-HETE in LD, however, the reduction was not statistically significant. This was added to the manuscript.

6. Extended figure 4a. Difference in limb diameter is marginal.

We agree with reviewer 3, the difference of limb diameter observed after PD146 is small, but it is strongly reproducible. Importantly, we observed a significant effect on other lymphedema hallmarks such as fibrosis, dermal backflow, and decrease in Treg cell number in mice treated with the inhibitor as described in Extended data Figure 4b,c,d and e.

7. Extended figure 4f. The flow plot is very hard to see. No numbers were shown on the gate to indicate frequencies of cells within the gate.

Representative image plots from Diva software were analyzed on Flowjo software for a better visibility and numbers were added to the gate according to reviewer 3 instructions.

8. Extended figure 4g, not sure what the authors mean by CD4+ cells (%CD4+). % CD4+ T cells among what cells?

We apologize for the mislabeling of the graph. Figure 4g means CD4+cells among CD3+cells. Legend was corrected.

9. Figure 3d. Again, no number of the gates to indicate frequency of cells in each gate.

Numbers are now indicated in the gate

Is the frequency/percentage of Tregs also reduced in LD? What about changes in other immune cells (Tconv, CD8+ T cells, macrophages, DCs?).

The percentage and the number of Treg cells is reduced in lymphedema tissue as shown in Figure 3d and e. According to reviewer instructions, we performed immune cell analysis in LD limb AT (Extended data Fig. 4a-j). F4/80-positive macrophages, M1 and M2 macrophage subpopulations, dendritic cells, CD8+ and CD4+ T cell populations were analyzed by flow cytometry. No difference was found in any of these cell population except for CD4+ T cell number that increases in LD.

10. No data showing efficiency and specificity of Alox15 KO following tamoxifen injection.

Staining Alox15-Lyve1 was added to extended fig. 6b to show the knock down of ALOX15.

11. Figure 3h, some dots were shifted to the right of the plot, maybe during figure editing.

We apologize for the figure editing, graph was corrected.

12. Figure 3i. Percent of foxp3+PPARg+ cells among what cells?

These data are from histological analysis; it is a percentage of positive cells per field.

13. Figure 4j, CCL21 and LTBR are reduced upon siALOX15 knockdown in primary cultures of HDLEC. Are these changes validated in ALOX15 conditional KO mice? More importantly, is this responsible for the reduction in Treg recruitment?

RT-qPCR CCL21 and LTBR were performed on conditional KO. CCL21 and LTBR expression is reduced in LD. However, no significant decrease was observed in knock out mice compared to control.

Additionally, using in vivo assay, we found that the increased number of Treg might be attributed to an inhibition of apoptosis, but not an increase of lymphatic transendothelial migration of adhesion.

REVIEWER COMMENTS

Reviewer #1 (Remarks to the Author):

The authors have thoughtfully and thoroughly addressed the prior concerns of the reviewers. I consider this manuscript to now be suitable for publication in Nature Communications.

Reviewer #2 (Remarks to the Author):

The manuscript is significantly improved. However, some additional edits will enhance its merit even further.

1. The new data in Ext. Fig 8 is very important and should be added to the main figures and indicated in the abstract. In addition, the authors should examine the indices indicated in panels g-l, as well as the expression of 12/15-LO (or other LOs), 15-HETE production and the number of Tregs, following treatment with IFN- β . This was not shown in the figure and the text in the results section (lines 227-230) is confusing. If data is commensurate with an involvement of IFN- β in lipoxygenases expression, 15-hete production and Treg numbers than this rescue effect should be added to the title. The images in panels e-f should be quantified in a graph.
2. The fact that satiated macrophages express IFN- β and 12/15-LO as exclusive markers during the resolution of inflammation (PMID: 31375662), and that these macrophages also express an anti-fibrotic gene signature (PMID: 32296415) is important in the context of this manuscript and should be explicitly indicated.
3. In Ext. Fig 3 the data presentation should be improved as previously requested- numbers added to heatmaps in d-e. data point should explicitly be shown in the other panels.
4. Comments that were not addressed: In Fig. 4d and ext Fig. 4b quantification of collagen content, reduced S.C. fat tissue thickness and number of hair follicles would strengthen the enhanced fibrosis conclusion.
5. In line 36 chemoattraction should be omitted, line 42 – invasion should be replaced with presence.
6. Anova should be corrected to ANOVA throughout.
7. Line 116- PMN and neutrophils are parallel. Omit one of them.

8. In line 134- change to “a reduced molecular weight form of 15-LO”.

9. Line 234- replace chemoattractive with "the".

Reviewer #3 (Remarks to the Author):

The authors have addressed many of the concerns that I raised, especially regarding the function of Tregs in LD with the adoptive transfer model, and dissecting whether LD influence the survival, proliferation, or recruitment of Tregs. Overall, the manuscript is significantly improved. However, there are still a few issues related to my original comments that remain to be addressed:

1. There are still a few places where the authors claimed that LD or ALOX15 affect recruitment of Tregs even though the authors stated in the rebuttal that they did not have data to support that claim: line 36 “chemoattraction”, line 83 “recruitment”, line 208 “recruitment”. Please either provide data to support their claim or change their claim in those parts.

2. Extended data Fig 1a, the PCA analysis showed that there are huge variabilities between individuals and Ctl and LD samples are not separated by groups. This does not support their claim that there are similar gene expression regulation patterns. They need to discuss what might have caused such variability and what steps they took to identify consistent gene expression changes between the two groups.

3. Figure 4g and 4h. How can there be more PPARg+ Tregs than total Tregs?

4. Differences between groups in many figures are very small (Fig 3b, extended Fig 5a, 5g, 5h, extended Fig 7h). They might be statistically significant, but are such small changes biologically significant? For example, in extended Fig 7h, the % of apoptotic Tregs reduces from 93% to 89% with 15-HETE. In extended Fig 5a, the limb diameter increased from 7.5mm to 7.8mm. These data were used to make several major claims in the paper but are such minor changes biologically meaningful?

REVIEWER

COMMENTS

Reviewer #1 (Remarks to the Author):

The authors have thoughtfully and thoroughly addressed the prior concerns of the reviewers. I consider this manuscript to now be suitable for publication in Nature Communications.

We thank reviewer 1 for supporting our work.

Reviewer #2 (Remarks to the Author):

The manuscript is significantly improved. However, some additional edits will enhance its merit even further.

We thank reviewer 2 for the careful evaluation of the manuscript. We made changes as requested below.

1. The new data in Ext. Fig 8 is very important and should be added to the main figures and indicated in the abstract. In addition, the authors should examine the indices indicated in panels g-l, as well as the expression of 12/15-LO (or other LOs), 15-HETE production and the number of Tregs, following treatment with IFN- β . This was not shown in the figure and the text in the results section (lines 227-230) is confusing. If data is commensurate with an involvement of IFN- β in lipoxygenases expression, 15-hete production and Treg numbers than this rescue effect should be added to the title. The images in panels e-f should be quantified in a graph.

The beneficial effect of IFN- β in lymphedema is now mentioned in the title and indicated in the abstract. Data are presented as main figure 5.

The expression of 12/15LO was investigated in response to IFN- β treatment in ALOX15^{LECKO} mice and their control littermates. We confirmed the downregulation of 12/15-LO in control mice that undergo through lymphedema (new Fig. 5c). Importantly, we found that this was reversed by IFN- β treatment that induced a reduction in collateral formation called “rerouting lymphatics” in the limb, a major hallmark of lymphedematous limb. Quantification is now provided in Figure 5e.

We found that IFN- β rescued Treg cell number in the LD limb form ALOX15^{lecko} mice (Figure 5h).

Additional quantification of CCL21 and LTBR gene expression was performed in IFN- β -treated mice (Figure 5i and j). The downregulation of CCL21 and LTBR observed in both ALOX15^{LECKO} mice and control littermates was abrogated by IFN- β treatment. This was previously observed as IFN- β increases CCL21 synthesis by lymphatic endothelial cells (Cufi, J autoimmune 2014) as mentioned in the discussion.

2. The fact that satiated macrophages express IFN- β and 12/15-LO as exclusive markers during the resolution of inflammation (PMID: 31375662), and that these macrophages also express an anti-fibrotic gene signature (PMID: 32296415) is important in the context of this manuscript and should be explicitly indicated.

We totally agree with reviewer 2 and we appreciate his contribution for highlighting the therapeutic properties of IFN- β in secondary lymphedema. It will definitely open new therapeutic opportunities for that unmet medical need.

Even if the current manuscript shows for the first time that resolution of inflammation plays a crucial role in lymphedema and is associated with the control of the Treg cell population function, a paragraph was added to the discussion so as not to neglect the potential role of macrophages in the resolution phase:

“Although no significant changes in PMN was observed in lymphedema, we cannot exclude some contribution of macrophages in the resolution phase. In particular, satiated or non-phagocytic macrophages exhibit distinct gene expression profile involved in tissue repair and express high levels of IFN β (Butenko, Front Immunol 2020). In particular, resolution phase macrophages express a selective IFN- β -related gene signature in mice model of bacterial inflammation (Kumaran, Nature Com 2019). In this model, treatment with exogenous IFN- β enhanced bacterial clearance, demonstrating that IFN- β produced by resolution phase macrophages is an effector cytokine in resolving bacterial inflammation. Also, IFN- β enhances clearance of apoptotic PMN via efferocytosis that is essential for prevention of chronic inflammation and autoimmunity (Korns, Front. Immunol. 2011). Therefore, IFN- β may represent a promising therapeutic target to restore the resolution of inflammation in lymphedema”

3. In Ext. Fig 3 the data presentation should be improved as previously requested- numbers added to heatmaps in d-e. data point should explicitly be shown in the other panels.

The lipidomic dosages from human (skin and adipose tissue) and mice lymphedematous tissues (pg/mg tissue) are now provided in supplementary materials in Table 2-10:

Table 2: Dosage of AA-derived lipids in human lymphedema skin tissue biopsies.

Table 3: Dosage of DHA-derived lipids in human lymphedema skin tissue biopsies.

Table 4 : Dosage of EPA-derived lipids in human lymphedema skin tissue biopsies.

Table 5: Dosage of AA-derived lipids in human lymphedema adipose tissue biopsies.

Table 6: Dosage of DHA-derived lipids in human lymphedema adipose tissue biopsies.

Table 7: Dosage of EPA-derived lipids in human lymphedema adipose tissue biopsies.

Table 8: Dosage of AA-derived lipids in mice lymphedema adipose tissue biopsies 2 and 8 weeks after surgery.

Table 9: Dosage of DHA-derived lipids in mice lymphedema adipose tissue biopsies 2 and 8 weeks after surgery.

Table 10: Dosage of EPA-derived lipids in mice lymphedema adipose tissue biopsies 2 and 8 weeks after surgery.

Data points are shown in Extended Figure 3 f and g.

4. Comments that were not addressed: In Fig. 4d and ext Fig. 4b quantification of collagen content, reduced S.C. fat tissue thickness and number of hair follicles would strengthen the enhanced fibrosis conclusion.

Quantification of subcutaneous adipose tissue and collagen is now provided in Figure 4 (lentivector ALOX15) and Extended figure 5 (PD146176 inhibitor treatment).

In the opposite of skin inflammation, no reduction of fat tissue thickness was observed in LD as it is a lymphatic disease characterized by an accumulation of fluid and adipose tissue in the limb followed by a chronic inflammation induced by the lymph stasis.

Hair follicles were quantified in both experiments. We observed a downregulation of hair follicles in LD fibrous skin. Interestingly, this was aggravated with 15-LO inhibitor (PD146176) and rescued with ALOX15 lentivector as shown now in Extended Fig. 5 and Figure 4 respectively.

Data are now discussed in the manuscript:

“We observed that LD skin fibrosis was associated with a downregulation of the number of hair follicles. This phenotype was previously described by Butenko and colleagues during acute inflammation in a mice model deficient for atypical chemokine receptor AKCR2 (Butenko Faseb j 2017). They found that degeneration and loss of hair follicles was associated with an augmented thickness of the collagenous dermis. Importantly, this was abrogated by IFN- β treatment. However, these observations were only associated with the dermis phenotype as LD is characterized by an increase of hypodermis AT that is not observed in skin inflammatory models.”

5. In line 36 chemoattraction should be omitted,

Correction was performed

line 42 – invasion should be replaced with presence.

Correction was performed

6. Anova should be corrected to ANOVA throughout.

Correction was performed

7. Line 116- PMN and neutrophils are parallel. Ommit one of them

Correction was performed

8. In line 134- change to “a reduced molecular weight form of 15-LO”.

Correction was performed

9. Line 234- replace chemoattractive with "the".

Correction was performed

Reviewer #3 (Remarks to the Author):

The authors have addressed many of the concerns that I raised, especially regarding the function of Tregs in LD with the adoptive transfer model, and dissecting whether LD influence the survival, proliferation, or recruitment of Tregs. Overall, the manuscript is significantly improved. However, there are still a few issues related to my original comments that remain to be addressed:

We thank reviewer 3 the encouraging feedback towards our manuscript. We hope that we have responded to the criticisms raised by reviewer 3.

1. There are still a few places where the authors claimed that LD or ALOX15 affect recruitment of Tregs even though the authors stated in the rebuttal that they did not have data to support that claim:

line 36 “chemoattraction”:

word was removed

line 83 “recruitment”:

Sentence was changed for: “decrease of T_{reg} cell number”

line 208 “recruitment”.

Sentence was changed for : “improved the number of $Foxp3+PPAR\gamma+$ cells “

Please either provide data to support their claim or change their claim in those parts.

2. Extended data Fig 1a, the PCA analysis showed that there are huge variabilities between individuals and Ctrl and LD samples are not separated by groups. This does not support their claim that there are similar gene expression regulation patterns. They need to discuss what might have caused such variability and what steps they took to identify consistent gene expression changes between the two groups.

We naturally agree with the reviewer's comments. There are indeed marked variations between individuals and Ctrl and LD samples. This variability in gene expression in a human population is normal and is the cumulative result of intrinsic genetic factors, variable penetrance of the disease, extrinsic environmental factors, and stochastic noise. The results we have chosen to present without correction reflect these variations and may appear confusing. However, this is only a deceptive appearance. Indeed, if we apply a batch effect correction of the inter-variabilities of the individuals through the DESEQ2 methods and if the effect of individual number 1 is blocked, the variabilities are strongly reduced and the two conditions LD and CTRL are well separated. Two figures supporting those statements are provided below, the batch corrected PCA and a heatmap showing clearly the separation between CTRL and LD for all individuals. We hope that the reviewer will share our belief that these new figures fully support our claims and conclusions."

Principal component analysis (PCA) plot, realized after the batch effect correction, showing clear separation by condition (CTRL versus LD). PC1 (43.48%) shows a significant difference between the samples between the CTRL and the LD conditions. PC2 (24,37%) separates the

control samples versus the pathological samples. Both axes explain 67.85 % of the data variability.

Hierarchical clustering heatmap of the differentially expressed genes between the CTRL and the LD condition of the RNA-seq data, after the batch effect of the individual n.1 was corrected.

3. Figure 4g and 4h. How can there be more PPARg+ Tregs than total Tregs?

We performed Foxp3 pixel quantification for Fig 4g, however, figure 4h express the number of PPARg positive Treg cells. Clarification was made in the figure legend.

4. Differences between groups in many figures are very small (Fig 3b, extended Fig 5a, 5g, 5h, extended Fig 7h). They might be statistically significant, but are such small changes biologically significant? For example, in extended Fig 7h, the % of apoptotic Tregs reduces from 93% to 89% with 15-HETE. In extended Fig 5a, the limb diameter increased from 7.5mm to 7.8mm. These data were used to make several major claims in the paper but are such minor changes biologically meaningful?

We agree with reviewer, there is little difference, however these are highly reproducible and difference in limb diameter between Ctrl and LD is clearly visible on mice before any measurement. Importantly, to make sure that the effect is relevant related to lymphedema, these measurements were associated to the other hallmarks of lymphedema: skin fibrosis and increased thickness, rerouting (collateral formation) of limb lymphatic vessels, and changes in immune Tcells populations. We have claimed a biological effect when the entire hallmarks were modified in the mouse model.

In Extended Fig 5a, despite a low effect of 15-LO inhibitor in limb diameter, we observed a strong effect in dermis thickness, a severe dermal lymph backflow with collateral formation and decrease in Treg cell number.

For the quantification of apoptosis in Extended Figure 7, it was an in vitro experiment that support the stronger data observed in vivo and presented in main figure.

REVIEWERS' COMMENTS

Reviewer #2 (Remarks to the Author):

The Manuscript is now acceptable for publication.

Reviewer #3 (Remarks to the Author):

The authors have addressed my comments. I would suggest that they include the PCA plot and heat map after batch correction (shown in the response to reviewer letter) in the revised manuscript.

NCOMMS-22-41943B
RESPONSE TO REVIEWERS' COMMENTS

Reviewer #2 (Remarks to the Author):

The Manuscript is now acceptable for publication.

We thanks reviewer 2 for the approval of the manuscript.

Reviewer #3 (Remarks to the Author):

The authors have addressed my comments. I would suggest that they include the PCA plot and heat map after batch correction (shown in the response to reviewer letter) in the revised manuscript.

The PCA plot and heat map after batch correction were included into the revised manuscript.